# Practical Certificate-Less Infrastructure with Application in TLS

**Li Duan** [1,2,†] [ID], **Yong Li** [1,*,†] and **Lijun Liao** [1,†]

1 Huawei Technologies Düsseldorf, 8600 Düsseldorf, Germany; li.duan@huawei.com (L.D.); lijun.liao@huawei.com (L.L.)
2 Department of Computer Science, Paderborn University, 33098 Paderborn, Germany
* Correspondence: yong.li1@huawei.com
† These authors contributed equally to this work.

**Abstract:** We propose highly efficient certificate-less (CL) protocols for the infrastructure used by authenticated key exchange (AKE). The construction is based on elliptic curves (EC) without pairing, which means it can be easily supported by most industrial cryptography libraries on constrained devices. Compared with other pairing-free CL solutions, the new CL-AKE protocol enjoys the least number of scalar multiplications over EC groups. We use a unified game-based model to formalize the security of each protocol, while most previous works only assess the security against a list of attacks, provide informal theorems without proper modeling, or use separate models for protocols in different stages. We also present an efficient integration of the core protocols into the TLS cipher suites and a stand-alone implementation for constrained devices. The performance is evaluated on constrained devices in real-world settings, which further confirms the efficiency of our proposal.

**Keywords:** certificate-less cryptography; authenticated key exchange; TLS; IoT security

## 1. Introduction

The authenticated key exchange protocols (AKE) based on conventional certificates are still widely deployed. Even in relatively new standards, such as TLS 1.3 [1], certificate-based cipher suites remains significant. Theoretical frameworks for evaluating protocol security, such as extensions of the Bellare–Rogaway model (BR) [2], eCK model [3] and universal composability (UC) [4], are usually used while assuming the existence of certificate-based public key infrastructure (PKI) [5,6].

However, the drawbacks of certificate-based infrastructure are also obvious. Ever-growing certificate revocation lists (CRL), floods of Online-Certificate States Protocol (OCSP) requests and the complicated logic of certificate chain verification may not fit into constrained devices [7] in the Internet of Things (IoT), where 50 KB RAM is already a luxury. Turning to pure symmetric key cryptography is not optimal in practice, either, as that introduces heterogeneity in the infrastructure, and scales poorly.

Therefore, exploring practical certificate-less AKE protocols (CL-AKE) is extremely meaningful.

CL-AKE solutions can use certificate-less public key encryption (CL-PKE) or signature (CL-SIG) to replace certificate chains. The syntax of the scheme and the types of adversaries have been defined for the first CL-PKE and CL-SIG by Al-Riyami and Paterson [8]. Unlike the key generation center (KGC) in identity-based or attribute-based cryptography (IBC, ABC), the KGC in CL-PKE can only compute partial private keys of users, so solutions based on CL-PKC do not suffer from the key escrow problem. The initial construction of CL-PKC in [8] is bilinear pairing based. Later, in 2007, Crampton et al. proposed a password-enabled and certificate-free grid security infrastructure (PECF-GSI) [9]. The protocols in PECF-GSI use bilinear pairing as heavily as the original Al-Ruyami–Paterson schemes. Other pairing-based certificate-less solutions in the last two decades [10–15] have experienced various improvement and trade-offs.

One of the major challenges for building certificate-less infrastructure is to optimize the efficiency of every protocol in its cryptographic core. As pointed out in [16], a bilinear pairing operation is about ten-times slower than a point multiplication on elliptic curves (EC), so it is necessary to avoid pairing and also to trim the redundant components, such as signature generation or decryption, off the CL-AKE main protocol.

Another critical challenge is establishing a unified security model for the two stages: user key registration and AKE. An adversary $\mathcal{A}$ against CL-PKE/SIG with key registration [8] has the power to corrupt users, corrupt KGC and register new public keys. However, whether $\mathcal{A}$ can see any message exchanged between an honest user and the KGC is undefined. In contrast, adversaries against AKE protocols have different powers in BR [2], eCK [3] and other game-based models [5]. These adversaries can tamper with messages and corrupt parties but may not be allowed to register new public keys. Although it has not been theoretically ruled out that messages in the key registration phase can threaten the AKE phase, most previous works on CL-AKE [15,17–19] used separate game-based models for the generation/registration of user key pairs (in a secure channel or out of band channel) and AKE protocol (in the public channel), or ignore the messages exchanged during the registration.

### 1.1. Our Contribution and Paper Outline

To meet the challenges mentioned above, we make the following contribution in this paper.

1.  We propose a practical cryptographic core of a certificate-less (CL) infrastructure, including user key registration and CL-AKE. The protocols are constructed from elliptic curves (EC) without pairing or any signature so that they can be easily supported by most industrial public key cryptography libraries for constrained devices. To the best of our knowledge, our AKE protocol also enjoys the optimal number of point multiplication over EC compared to other pairing-free solutions (see Table 1).
2.  We integrate CL-AKE into TLS ciphersuites [1]. The performance is compared with TLS-DHE with certificates in data volume and computation. We also deploy and test the slim implementation of CL-AKE without the TLS stack on constrained IoT devices. Subsequently, the evaluation confirms the real-world efficiency of our proposal.
3.  Our new provably secure CL signature scheme $\Pi_{\mathsf{CL\text{-}SIG}}$ with two-way public key reconstruction can be of independent interest.

**Table 1.** Comparison with other <u>provably secure</u> pairing-free *(EC)DH-based* CL-AKE. Proposals without security models, such as [18,20], are not included. **BPM**: base-point multiplication on each side; **PM**: non-base-point multiplication on each side;

|  | # BPM | # PM | Security Model |
|---|---|---|---|
| Yang and Tan [17] | 1 | 10 | dedicated, game-based, stage-separated |
| Song et al. [16] | 1 | 7 | dedicated, game-based, stage-separated |
| He et al. [19] | 1 | 4 | eCK for CL-AKA only |
| **This work** | **1** | **3** | extended eCK for AKE and key reg. |

After the introduction, we present the necessary notation and preliminaries in Section 2. As a starting point for building CL-AKE and proving its security in the game-based framework, we introduce a new certificate-less and paring-free signature scheme with two-way public key reconstruction in Section 3. The game-based AKE security model is presented in Section 4. The new certificate-less key registration, CL-AKE protocols and the security analysis can be found in Section 5. We present the integration to TLS and the evaluation in Section 6.

*1.2. Technical Road Map*

The cost of verifying a conventional certificate chain is proportional to the number of certificates on the chain. More specifically, to verify the end-level signature, a user has to verify the first-level signature with the public key in the root certificate and then the second-level signature with the public key of the first-level certificate. The verification continues until the signature on the end-level certificate is verified. For EC-based signatures, the verification usually involves a significant amount of point multiplication in an EC group.

A CL-SIG is designed to identify and utilize shortcuts during the verification process. Ideally, a user only has to verify the end-level public key with the root public key, usually the public key of the KGC. This verification can also be enforced implicitly through computation. If an end-level public key $pk_j$ can be reconstructed by any honest user using the KGC's public key $pk_{KGC}$ and the identifier $PID_j$, then intuitively, the verification of $pk_j$ is almost finished.

The shortcut we implement in CL-SIG is to replace the signature verification with a hash function, which is considerably more efficient. The hash function $H_1()$ maps binary strings to elements in an integer group. If a public key is an EC point that can be encoded as a binary string, and the corresponding private key is an integer in a group, $H_1()$ provides an efficient way to bind the public keys $pk_j$, $pk_{KGC}$ and the identifier $PID_j$ with the private key. Moreover, to make CL-SIG fully functional, there must be **two** alternatives to how $pk_j$ can be reconstructed. One is through $sk_j$, i.e., the way that only the key owner can take, and another one is through the use of $H_1()$, $pk_{KGC}$ and some additional information $B_j$, i.e., any user can do it. For details, we refer the reader to Figure 1 in Section 3.

We implement another shortcut to construct CL-AKE from CL-SIG. Instead of using the signing and verification algorithms in our CL-SIG, we keep only the public key reconstruction algorithms in the AKE. An AKE participant can reliably reconstruct its peer's public key $pk_j$ and then use $pk_j$ with its own ephemeral key materials to derive the session keys. A message authenticate code is used to confirm the knowledge of all related secrets, replacing the expensive signature verification. For details, we refer the reader to Figure 2 in Section 5.

In summary, we replace as many public key operations (e.g., signature and point multiplication) with efficient symmetric essential operations (e.g., hash, message authentication code and pseudo-random function) as possible, while keeping the CL solution provably secure. The resulting CL-AKE enjoys forward secrecy due to the Gap Diffie–Hellman problem's hardness and the new CL-SIG's security.

*1.3. Related Work*

We review existing approaches to construct certificate-less AKE and authentication infrastructure, including identity-based cryptography, attribute-based cryptography, and CL-AKE with and without pairing.

1.3.1. IBC and ABC-Based CL Solutions

An important line of research is replacing certificate-based PKI with identity-based cryptography (IBC) [21]. In principle, the IBC public key is a user identity pid itself, and pid is embedded algebraically into the user's secret key by KGC with its master secret key msk. IBC eliminates certificates and simplifies the management of public keys greatly, but it suffers from the key escrow problem. More specifically, in the standard syntax of IBC, such as in [21,22], every user secret key is derived from its pid and a system-wide static msk of KGC. Once msk is compromised, the adversary can use msk to recover all previous user secret keys, destroying forward secrecy (FS). Attribute-based cryptography (ABC) [23] can be seen as a generalization of IBC. Instead of using one single identity, ABC uses a combination of multiple attributes to encrypt a message. However, the key escrow problem remains if any user secret key is derivable from msk and the attribute combinations alone. This KGC setting is preserved in various IBC-/ABC-based solutions [24,25], and some are flawed or without FS later [15,26].

### 1.3.2. CL-PKC and Pairing-Based Attempts

The notion of certificate-less public key cryptography (CL-PKC) was first formalized in 2003 by Al-Riyami and Paterson [8]. As the KGC in CL-PKC can only compute partial private keys for users, solutions based on CL-PKC do not inherently suffer from the key escrow problem. Later, Crampton et al. proposed a password-enabled and certificate-free grid security infrastructure (PECF-GSI) [9] in 2007. The protocols in PECF-GSI use bilinear pairing heavily as the original Al-Ruyami–Paterson schemes.

Various pairing-based attempts have been made for different trade-offs between security and efficiency. In 2012, Sanaa Taha et al. proposed certificate-less authentication key agreement (CL-AKA), a link-layer authentication and key agreement protocol based on CL-PKC, which does not consider ephemeral key leakage attacks [10]. Maity et al. proposed a novel certificate-less on-demand public key management (CL-PKM) protocol for self-organized MANETs [27]. Memon et al. proposed two authentication protocols based on Al-Ruyami–Paterson CL-PKC and IBE [11,12] in 2015. The security is analyzed with BAN-logic. Balakrishnan et al. proposed a practical email system based on CL-PKC with user authentication and key exchange in 2016, but the encryption of messages actually bears no forward secrecy when the receiver's long-term keys are exposed [13]. Bala et al. proposed a secure key management and authentication protocol in 2017, making use of hybrid cryptography that involves both symmetric and CL-PKC but without formal security models [14]. Saeed et al. proposed a lightweight online/offline certificate-less signature (L-OOCLS) and a heterogeneous remote anonymous authentication protocol (HRAAP) for IoT applications in 2018 [15]. The L-OOCLS scheme is pairing-based and provably secure in random oracle model. The proposed HRAAP, however, cannot be proved secure in the BR or eCK model as the session key is directly used in handshake.

### 1.3.3. Pairing-Free CL-AKE

Pairin-free CL solutions have also been proposed. Song et al. [16] proposed a secure lightweight certificate-less authenticated key agreement (CL-AKA) for securing vehicle-to-vehicle (V2V) communication without using pairings. Unfortunately, the protocols need a large number of exponentiations in an integer group. Yang and Tan [17] proposed a CL-AKA that is provably secure in a dedicated model. He et al. [19] proposed efficient CL-AKA with security proofs in an extended eCK model dedicated to the key agreement part alone. Farouk et al. also introduced an efficient pairing-free CL-AKA protocol for grid computing environments [18] by extending the work of He et al. [19]. In 2018, KhanSafi et al. proposed an authentication framework for the message dissemination of toll payment information with a pairing-free CL-PKC system [20]. Unfortunately, there is no security proof provided in [19,20].

Defining a unified security model for each stage of CL-AKE is not a trivial task. On the one hand, the adversary for CL-PKE or CL-SIG in [8] has the power to corrupt users, corrupt the KGC or register new public keys, but cannot see any messages exchanged between the user and KGC. On the other hand, adversaries against authenticated key exchange (AKE) protocols, however, have different powers in BR [2], eCK [3] and other game-based models [5]. These adversaries can tamper with messages and corrupt parties (users) but cannot register new public keys. However, it has *not* been confirmed or denied whether messages in the key pair generation phase can threaten the AKE phase. Most previous works on CL-AKE [10,15,16,18,19] used separate models for the generation of user key pairs (in a secure channel or out-of-band channel) and AKE protocol (in public channels), or even ignore the messages exchanged during the key pair generation.

From 2021 till now, lattice-based (LBC) and isogeny-based cryptography have been introduced for post-quantum security, and new constructions have been proposed [28–32]. However, deploying LBC on constrained devices remains challenging now and in the near future, especially when facing the conflict between the large key/ciphertext size required by LBC and the limited RAM/storage on constrained devices [33].

## 2. Notation and Preliminaries

In this section, we introduce the necessary cryptographic building blocks of our solution.

### 2.1. Notations

We use $\kappa \in \mathbb{N}$ and $1^\kappa$ to denote the security parameter. Let $[n] = \{1, \ldots, n\} \subset \mathbb{N}$ be the set of integers from 1 to $n$. If $S$ is a set, $a \xleftarrow{\$} S$ means sampling a uniformly random element $a$ from $S$. If $\mathcal{A}()$ is an algorithm, $m \leftarrow \mathcal{A}^{\mathcal{O}(\cdot)}(x)$ and $\mathcal{A}^{\mathcal{O}(\cdot)}(x) \xrightarrow{\$} m$ denote that $\mathcal{A}$ outputs $m$ on input $x$ with the help of another oracle $\mathcal{O}(\cdot)$. $X||Y$ means concatenating two binary strings $X$ and $Y$. We use $\Pr[E : A]$ to denote the probability that event $E$ happens if action $A$ is taken. Other notations will be introduced as needed.

### 2.2. Cryptographic Primitives and Hardness Assumptions

Message authentication code (MAC) is frequently used in AKE protocols for message integrity and can also work as a proof of the knowledge of the secret key.

**Definition 1** (Message Authentication Code, MAC). *A MAC scheme* MAC $=$ (MAC.Gen, MAC.Tag, MAC.Vfy) *consists of three algorithms:* MAC.Gen, MAC.Tag *and* MAC.Vfy *described below.*

- MAC.Gen$(1^\kappa) \xrightarrow{\$}$ k. *The non-deterministic key generation algorithm* MAC.Gen$()$ *takes the security parameter* $1^\kappa$ *as the input and outputs the secret key* k.
- MAC.Tag$(k, m) \xrightarrow{\$}$ mTag. *The (non-deterministic) message tagging algorithm* MAC.Tag$()$ *takes the secret key* mTag *and a message* m *as the input and outputs the authentication tag* mTag.
- MAC.Vfy$(k, m, \text{mTag}) = b$. *The deterministic tag verification algorithm* MAC.Vfy$()$ *takes the MAC secret key* k, *a message* m *and a tag* mTag *as input and outputs a boolean value* b. b *is* TRUE *if* mTag *is a valid MAC tag on* m.

Hash functions are used for obtaining a digest of the input. The digest can be of fixed length or in a finite domain.

**Definition 2** (Collision-resistant Hash Function). *A hash function* $H : \mathcal{M} \rightarrow \mathcal{D}$ *is collision-resistant if there exists a negligible function* $\epsilon_{\text{coll}}()$ *such that for any algorithm* $\mathcal{A}$ *with running time bounded by* $\text{poly}(\kappa)$, *it holds that*

$$\Pr\begin{bmatrix} (m_0, m_1) \leftarrow \mathcal{A}(1^\kappa, H) : \\ m_0 \neq m_1 \wedge H(m_0) = H(m_1) \end{bmatrix} \leq \epsilon_{\text{coll}}(\kappa),$$

*where* $\mathcal{M}$ *is the message space, and* $\mathcal{D}$ *is the hash image space.*

Pseudo-random function (PRF) can be used for key derivation as in TLS 1.3 [1]. PRF ensures that the output looks random if the secret key is not leaked.

**Definition 3** (Pseudo-random function, PRF). *A pseudo-random function* $\mathbb{F} = (\text{FKGen}, \text{PRF})$ *consists of two algorithms,* FKGen *and* PRF, *described below.*

- FKGen$(1^\kappa) \xrightarrow{\$}$ k. *The non-deterministic key generation algorithm* FKGen$()$ *takes the security parameter* $1^\kappa$ *as the input and outputs the secret key* k.
- PRF$(k, x) = y$. *The PRF evaluation algorithm* PRF$()$ *takes as the input the secret key* k *and a value* x *in the domain and outputs an image* y.

The Schnorr signature scheme can be seen as a general template for (EC-)group-based signature. The most critical operation is the scalar-point multiplication. Note that the verification algorithm SIG.Vfy of Schnorr needs two point multiplication, one on the base point $G$ and one on the non-base point pk. In contrast, the signing only needs one base

point multiplication. In practice, base point multiplication has been optimized for each EC group, so it is usually much quicker than non-base point multiplication.

**Definition 4** (Schnorr signature scheme)**.** *Let* $H_2() : \{0,1\}^\kappa \to \mathbb{Z}_q$ *be a collision-resistant cryptographic hash function. The Schnorr signature scheme* SIG *consists of three algorithms* (SIG.Gen, SIG.Sign, SIG.Vfy) *described below.*

- SIG.Gen$(1^\kappa) \xrightarrow{\$}$ (params, pk, sk)*. The non-deterministic key generation algorithm* SIG.Gen() *takes the security parameter* $1^\kappa$ *as the input and outputs the public parameters* params, *the public key* pk *and the corresponding private key* sk, *where* params $= (\mathbb{G}, G, H_2())$, *G is the generator of group* $\mathbb{G}$ *of large prime order q,* pk $= x \cdot G$, sk $= x$ *with* $x \xleftarrow{\$} \mathbb{Z}_{|\mathbb{G}|}$, *and* H() *maps any bit string to an integer in* $\mathbb{Z}_q$.
- SIG.Sign$($sk$, m) \xrightarrow{\$} \sigma$*. This signing algorithm* SIG.Sign() *takes the private key* sk *and the message m as the input. It chooses* $r \xleftarrow{\$} \mathbb{Z}_q$, *computes* $R = r \cdot G$, $e = H_2(R||m)$, *and* $\beta = r + e \cdot$ sk*. It outputs the signature* $\sigma = (R, \beta)$.
- SIG.Vfy$($pk$, m, \sigma) = b$*. This verification algorithm* RingVrfy() *takes a public key* pk, *a message m and a signature* $\sigma = (R, \beta)$ *as input. It first computes* $e' = H_2(R||m)$, *then outputs* TRUE *if* $\beta \cdot G = R + e' \cdot$ pk, *and* FALSE *otherwise.*

We refer the reader to standard cryptography literature, such as [34], for the security definition of all the cryptographic primitives above and Diffie–Hellman key exchange (DH).

**Definition 5** (Discrete logarithm, DL)**.** *Let* GGen$(1^\kappa)$ *be a group generation algorithm which outputs* params $= (\mathbb{G}, G, q)$, *where* $\mathbb{G}$ *is the description of a cyclic group, with G as its generator and q as its order. The discrete logarithm (DL) assumption with respect to* $\mathbb{G}$ *states that the following quantity is negligible for any probabilistic polynomial time (PPT) adversary* $\mathcal{A}$.

$$\mathsf{Adv}_{\mathcal{A},\mathsf{DL}} := \Pr\left[Y = x \cdot G : Y \xleftarrow{\$} \mathbb{G}; x \leftarrow \mathcal{A}(\mathsf{params}, Y)\right]$$

The proof of Schnorr's security or schemes that use group elements with hash usually relies on the hardness of the Discrete logarithm problem above. For proving security of AKE, we need the gap computational Diffie–Hellman Problem (GCDH), which is defined as: given public parameter params and $(a \cdot G, b \cdot G)$ for $a, b \in \mathbb{Z}_q$, compute the element $Z = (ab) \cdot G$ with the help of a Decisional Diffie–Hellman Oracle $\mathcal{O}_{\mathsf{ddh}}(\cdot)$, i.e., $\mathcal{O}_{\mathsf{ddh}}(\cdot)$ answers whether a given quadruple $(G, a \cdot G, b \cdot G, c \cdot G)$ has $ab \equiv c \mod q$.

**Definition 6** (Hardness of GCDH)**.** GCDH *is hard with respect to* $\mathbb{G}$, *if for any PPT adversary* $\mathcal{A}$, *the following quantity is negligible.*

$$\mathsf{Adv}_{\mathcal{A},\mathsf{GCDH}} := \Pr\left[Z = (ab) \cdot G : a, b \xleftarrow{\$} \mathbb{Z}_q; Z \leftarrow \mathcal{A}^{\mathcal{O}_{\mathsf{ddh}}(\cdot)}(\mathsf{params}, a \cdot G, b \cdot G)\right]$$

In this section, we have reviewed the most relevant cryptographic primitives and hard problems. In the next section, we show how to construct an efficient CL-SIG from them.

## 3. New Certificate-Less Signature with Two-Way Reconstructable Public Key

We construct an extended certificate-less signature (CL-SIG) as the starting point. Although signing and verification are not used in our CL-AKE protocol, the security of $\Pi_{\mathsf{CL\text{-}SIG}}$ simplifies the argument in the game-based framework.

**Definition 7** (CL-SIG with Two-way Reconstructable PK)**.** *A certificate-less signature scheme with a two-way reconstructable public key (CL-SIG-TRK) is a tuple of seven algorithms* (Setup, PPKey-Extract, Set-Private-Key, Set-Secret-Value, Set-Public-Key, Sign, Verify, Reconst-Pk) *defined as follows.*

- Setup$(1^\kappa) \xrightarrow{\$} (\mathsf{params}, \mathsf{msk})$. *The (non-deterministic) algorithm* Setup$()$ *takes in the security parameter $1^\kappa$ and outputs the system parameters* params *and the master key* msk.
- Set-Secret-Value$(\mathsf{params}, \mathsf{PID}_i) \xrightarrow{\$} (a_i, A_i)$. *This algorithm outputs party $i$'s secret value $a_i$ and auxiliary information $A_i$ on input* params *and the identifier* $\mathsf{PID}_i$.
- PPKey-Extract$(\mathsf{params}, \mathsf{msk}, \mathsf{PID}_i, A_i) \xrightarrow{\$} (s_i, \boxed{B_i})$. *This partial key extraction algorithm outputs party $i$'s partial private key $s_i$ and the partial public key $B_i$ on input* params, msk, $\mathsf{PID}_i$ *and* $A_i$.
- Set-Public-Key$(\mathsf{params}, s_i, B_i, a_i) \rightarrow \mathsf{pk}_i$. *This algorithm takes as input* params, $s_i$, $a_i$ *and* $B_i$, *and outputs $i$'s public key* $\mathsf{pk}_i$.
- $\boxed{\text{Reconst-Pk}(\mathsf{params}, \mathsf{PID}_i, B_i) \rightarrow \mathsf{pk}_i}$. *This public key reconstruction algorithm takes as input* params, *identity* $\mathsf{PID}_i$ *and the partial public key $B_i$, and it outputs the complete public key $\mathsf{pk}_i$ of party $i$.*
- Sign$(\mathsf{params}, \mathsf{sk}_i, m) \xrightarrow{\$} \sigma$. *This algorithm takes* params, *the private signing key $\mathsf{sk}_i$ and a valid message $m$ as input and outputs a signature $\sigma$.*
- Verify$(\mathsf{params}, \mathsf{pk}_i, \mathsf{PID}_i, m, \sigma) \rightarrow b$. *This algorithm outputs a bit value $b \in \{$TRUE, FALSE$\}$ on input* $\mathsf{pk}_i$, $\mathsf{PID}_i$, $m$ *and a signature $\sigma$. The value $b$ is* TRUE *if $\sigma$ is a valid signature on $m$ with respect to $\mathsf{pk}_i$ and* $\mathsf{PID}_i$.*

Algorithms and outputs in dashed boxes are the extensions to the syntax in [8]. In the original syntax, $\mathsf{pk}_i$ can only be computed by its owner with Set-Public-Key$()$. The extension Reconst-Pk$(\mathsf{params}, \mathsf{PID}_i, B_i)$, allows anyone who knows $B_i$ and the KGC's pubic key to reconstruct $\mathsf{pk}_i$. Thus, there are **two** ways to reconstruct the public key, giving space for more efficiency improvement in the CL-AKE construction.

The security game for CL-SIG in [8] is an EUF-CMA game extended with queries in Table 2. In principle, Type I adversaries can replace public keys but cannot get KGC's private key, while Type II adversaries can have the KGC's private key but cannot replace public keys.

**Table 2.** Queries (adversary's ability) in the CL-SIG security game [8]. Notations are adapted to ours.

| Query | Description |
|---|---|
| PPKey-Extract$(\mathsf{PID}_i)$ | return partial private/public keys |
| ReplacePK$(\mathsf{PID}_i, \mathsf{pk}_i')$ | replace the public key of $\mathsf{PID}_i$ with $\mathsf{pk}_i'$ |
| getPrivateKey$(\mathsf{PID}_i)$ | get the private key of $\mathsf{PID}_i$ |
| getKeyKGC$()$ | return msk |
| SIG.Sign$(\mathsf{PID}_i, m)$ | get $\mathsf{PID}_i$'s signature on $m$ |

More specifically, let $\mathsf{PID}_j$ be the challenged party and $(m^*, \sigma^*)$ the forgery. Besides being forbidden to ask getKeyKGC$()$, the restrictions on a Type I adversary $\mathcal{A}$ are :

1. $\mathcal{A}$ cannot query PPKey-Extract$(\mathsf{PID}_j)$.
2. For any $\mathsf{PID}_i$, $\mathcal{A}$ cannot query getPrivateKey$(\mathsf{PID}_i)$, if it has previously queried ReplacePK$(\mathsf{PID}_i, \mathsf{pk}_i')$.
3. $\mathcal{A}$ cannot query ReplacePK$(\mathsf{PID}_j, \mathsf{pk}_j')$ before submitting forgery, if it has previously asked PPKey-Extract$(\mathsf{PID}_j)$.
4. $\mathcal{A}$ has not queried SIG.Sign$(\mathsf{PID}_j, m^*)$ before submitting $(m^*, \sigma^*)$.

Besides being forbidden to ask ReplacePK$(\mathsf{PID}_i, \mathsf{pk}_i)'$ for any $i$, the restrictions on a Type II adversary $\mathcal{A}$ are :

1. $\mathcal{A}$ cannot ask getPrivateKey$(\mathsf{PID}_j)$.
2. $\mathcal{A}$ has not queried SIG.Sign$(\mathsf{PID}_j, m^*)$ before submitting $(m^*, \sigma^*)$.

Our new CL-SIG-TRK $\Pi_{\mathsf{CL\text{-}SIG}}$ is in Figure 1, where SIG is the Schnorr signature scheme in Definition 4. The security of $\Pi_{\mathsf{CL\text{-}SIG}}$ is summarized in Theorem 1. We still

stick to the original syntax of Verify(). On the other hand, the extension, Reconst-Pk(), also provides more flexibility in the verification, as a signature $(\sigma', B_i)$ can now be verified with itself and the KGC public key $\mathsf{pk}_{\mathsf{KGC}}$. We will show how a receiver of the partial public key $B_i$ can check and use it efficiently in AKE in Section 5.

Setup($1^\kappa$)

1 : Choose EC group $\mathbb{G}$ with base point $G$;

2 : $\mathsf{sk}_{\mathsf{KGC}} \xleftarrow{\$} \mathbb{Z}_{|\mathbb{G}|}; \mathsf{pk}_{\mathsf{KGC}} = \mathsf{sk}_{\mathsf{KGC}} \cdot G$;

3 : $\mathsf{params} = (\mathbb{G}, G, \mathsf{PID}_{\mathsf{KGC}}, \mathsf{pk}_{\mathsf{KGC}})$;

4 : $\mathsf{msk} = \mathsf{sk}_{\mathsf{KGC}}$;

5 : **return** $(\mathsf{params}, \mathsf{msk})$;

Set-Secret-Value($\mathsf{params}, \mathsf{PID}_i$)

1 : $a_i \xleftarrow{\$} \mathbb{Z}_{|\mathbb{G}|}, A_i = a_i \cdot G$;

2 : **return** $(a_i, A_i)$

PPKey-Extract($\mathsf{params}, \mathsf{msk}, \mathsf{PID}_i, A_i$)

1 : $b_i \xleftarrow{\$} \mathbb{Z}_{|\mathbb{G}|}, B_i = A_i + b_i \cdot G$;

2 : $s_i = b_i + \mathsf{H}_1(\mathsf{PID}_i||\mathsf{PID}_{\mathsf{KGC}}||B_i) \cdot \mathsf{msk}$;

3 : **return** $(s_i, B_i)$;

Set-Private-Key($\mathsf{params}, s_i, B_i, a_i$)

1 : $T_i = \mathsf{H}_1(\mathsf{PID}_i||\mathsf{PID}_{\mathsf{KGC}}||B_i) \cdot \mathsf{pk}_{\mathsf{KGC}}$

2 : **if** $s_i \cdot G \neq B_i - A_i + T_i$ : **return** $\perp$;

3 : $\mathsf{sk}_i = (a_i, s_i, T_i)$;

4 : **return** $\mathsf{sk}_i$;

Set-Public-Key($\mathsf{params}, s_i, B_i, a_i$)

1 : $\mathsf{pk}_i = (a_i + s_i) \cdot G$;

2 : **return** $\mathsf{pk}_i$

Reconst-Pk($\mathsf{params}, \mathsf{PID}_i, B_i$)

1 : $h = \mathsf{H}_1(\mathsf{PID}_i||\mathsf{PID}_{\mathsf{KGC}}||B_i)$;

2 : $T_i = h \cdot \mathsf{pk}_{\mathsf{KGC}}$;

3 : $\mathsf{pk}_i = B_i + T_i$;

4 : **return** $\mathsf{pk}_i$

Sign($\mathsf{params}, \mathsf{sk}_i, m$)

1 : Parse $\mathsf{sk}_i$ as $(a_i, s_i, T_i)$;

2 : $B_i = s_i \cdot G + a_i \cdot G - T_i$;

3 : $\mathsf{sk}'_i = a_i + s_i$;

4 : $\sigma' = \mathsf{SIG.Sign}(\mathsf{sk}'_i, m)$;

5 : $\sigma = (\sigma', B_i)$;

6 : **return** $\sigma$

Verify($\mathsf{params}, \mathsf{PID}_i, \mathsf{pk}_i, m, \sigma$)

1 : Parse $\sigma$ as $(\sigma', B_i)$;

2 : $h = \mathsf{H}_1(\mathsf{PID}_i||\mathsf{PID}_{\mathsf{KGC}}||B_i)$;

3 : $T_i = h \cdot \mathsf{pk}_{\mathsf{KGC}}$;

4 : **if** $\mathsf{pk}_i \neq B_i + T_i$ : **return** FALSE;

5 : **return** $\mathsf{SIG.Vfy}(\mathsf{pk}_i, m, \sigma')$;

**Figure 1.** Construction 1, the pairing-free CL-SIG-TRK construction $\Pi_{\mathsf{CL\text{-}SIG}}$. Capital letters, such as $A$, $B$ and $T$, represent EC group elements (points). Lowercase letters, such as $a$, $b$ and $s$, represent integers in $\mathbb{Z}_{|\mathbb{G}|}$.

**Theorem 1** (Security of $\Pi_{\mathsf{CL\text{-}SIG}}$). *If the discrete logarithm problem is hard with respect to group $\mathbb{G}$, then the CL-SIG-TRK scheme $\Pi_{\mathsf{CL\text{-}SIG}}$ is existentially unforgeable against chosen message attack (EUF-CMA) in the presence of Type I and Type II adversaries in the random oracle model, where Type I and Type II adversaries have access to queries defined in [8].*

*More specifically, if there exists $\mathcal{S}$ against $\Pi_{\mathsf{CL\text{-}SIG}}$, then there exist DL problem solvers $\mathcal{D}$ and $\mathcal{U}$, such that*

$$\mathsf{Adv}_{\mathcal{S},\mathsf{CL\text{-}SIG}} \leq 2d \cdot \mathsf{Adv}_{\mathsf{SIG}} + \sqrt[4]{(q_{\mathsf{H}_1} + q_{\mathsf{H}_2})^6 \cdot \mathsf{Adv}_{\mathcal{D},\mathbb{G}}^{\mathsf{DLP}}}$$
$$+ \sqrt[4]{\frac{3 \cdot (q_{\mathsf{H}_1} + q_{\mathsf{H}_2})}{|\mathbb{G}|}} + \frac{d^2 + q_{\mathsf{H}_1}^2 + 2 \cdot q_{\mathsf{H}_2}^2 + 2}{|\mathbb{G}|} \tag{1}$$

$$\textit{with } \mathsf{Adv}_{\mathsf{SIG}} \leq \sqrt[2]{(q_{\mathsf{H}_2} + q_{\mathsf{SIG}} + 1) \cdot \mathsf{Adv}_{\mathcal{U},\mathbb{G}}^{\mathsf{DLP}}}$$
$$+ \sqrt[2]{\frac{(q_{\mathsf{H}_2} + q_{\mathsf{SIG}} + 1) \cdot (q_{\mathsf{SIG}} + 1)}{|\mathbb{G}|}},$$

*where* $\mathsf{Adv}_{\mathcal{S},\mathsf{CL\text{-}SIG}}$ *is the advantage of any PPT adversary* $\mathcal{S}$ *against* $\Pi_{\mathsf{CL\text{-}SIG}}$, $\mathsf{Adv}_{\mathcal{D},\mathbb{G}}^{\mathsf{DLP}}$ *and* $\mathsf{Adv}_{\mathcal{U},\mathbb{G}}^{\mathsf{DLP}}$ *the advantage of* $\mathcal{D}$ *and* $\mathcal{U}$ *against* DLP, *respectively*, $\mathsf{Adv}_{\mathsf{SIG}}$ *the advantage of any PPT adversary against Schnorr Signature* SIG, *d the maximal number of clients with distinct identifiers*, $q_{\mathsf{H}_1}$ *the number of queries to random oracle* $\mathsf{H}_1$ *used in* $\Pi_{\mathsf{CL\text{-}SIG}}$, $q_{\mathsf{H}_2}$ *the number of queries to random oracle* $\mathsf{H}_2$ *used in Schnorr* SIG, *and* $q_{\mathsf{SIG}}$ *the number of signing queries.*

We defer the proof of Theorem 1 to Appendix A, as it relies on the multiple forking lemmma in [35] and is rather technical. Intuitively, the multiple-forking lemma helps us connect the CL-SIG security to the hardness of DLP (Definition 5).

From Theorem 1, we can have Corollary 1, which is necessary for proving the security of the user key registration protocol (see Figure 3 in Section 5). The proof of Corollary 1 is quite straight forward, as unforgeable CL-SIG implies unforgeable and non-replaceable key pairs.

**Corollary 1.** *If the advantages of Type I and Type II adversaries against* $\Pi_{\mathsf{CL\text{-}SIG}}$ *are upper-bounded by* $\mathsf{Adv}_{\mathsf{CL\text{-}SIG}}$, *the probability that any public-private key pair* $(\mathsf{pk}_i, \mathsf{sk}_i)$ *is forgeable or replaceable by Type I and Type II adversaries is also upper-bounded by* $\mathsf{Adv}_{\mathsf{CL\text{-}SIG}}$.

Now, we have all the necessary tools to construct a CL-AKE with forward secrecy [1].

## 4. Game-Based Security Model for CL-AKE

In this section, we define a security model for certificate-less AKE protocols. We assume each participant communicates through a public network, and the adversary controls all the data traffic. This setting is formalized in the execution environment.

### 4.1. Protocol Execution Environment

Let $\mathcal{SK} \in \{0,1\}^\kappa$ denote the session key space, and $\mathcal{K}$ is the pre-shared key space. Let $\{P_1, \ldots, P_\ell\}$ be the set of all parties for $\ell \in \mathbb{N}$, where a potential participant $P_i$ has a long-term pre-shared key $\mathsf{K} \in \mathcal{K}$ that corresponds to its identity $i$.

Each $P_i$ can have a polynomial number of process oracles $\{\pi_i^s\}$, where $s \in [d]$ is an index with $d \in \mathbb{N}$. A **unique** session identifier sid labels a protocol session between a client and a server instance. Moreover, we assume that besides the access to long-term secrets, such as private keys, each oracle $\pi_i^s$ maintains a list of independent internal state variables as described in the following list (Table 3).

**Table 3.** Internal states of oracles.

| Variable | Description |
| --- | --- |
| $\mathsf{PID}_i^s$ | records the identities $\{j\} \subset \{1, \ldots, \ell\}$ of intended communication partners $\{P_j\}$ |
| $\Phi_i^s$ | denotes $\Phi_i^s \in \{\texttt{accept}, \texttt{reject}\}$ |
| $\mathsf{sid}_i^s$ | denotes the session identifiers |
| $\mathsf{K}_i^s$ | records the session key $\mathsf{K}_i^s \in \mathcal{K}$ |
| $\mathsf{Eph}_i^s$ | records the ephemeral secret used to compute the session key $\mathsf{K}_i^s$ |

The internal state of each oracle $\pi_i^s$ is initialized as $(\mathsf{PID}_i^s, \Phi_i^s, \mathsf{sid}_i^s, \mathsf{K}_i^s, \mathsf{Eph}_i^s) = (\varnothing, \varnothing, \varnothing, \varnothing, \varnothing)$, where $\varnothing$ denotes the empty string. We assume that the session key is assigned to the variable $\mathsf{K}_i^s$ such that $\mathsf{K}_i^s \neq \varnothing$ if each oracle completes the execution with an internal state $\Phi_i^s = \texttt{accept}$.

### 4.2. Adversary Model

An active adversary $\mathcal{A}$ can interact with the execution environment by issuing the queries below. Queries in the dashed boxes are our extensions to the eCK model [3].

- $\mathsf{Send}(\pi_i^s, m)$: $\mathcal{A}$ can use this query to send any message $m$ of its choice to oracle $\pi_i^s$. The oracle will respond according to the protocol specification and its internal state. If

$m$ consists of a special symbol $\top$ ($m = \top$), then $\pi_i^s$ will respond with the first protocol message.

- RegCorruptParty($i, \mathsf{sk}_i, \mathsf{pk}_i$) This query allows $\mathcal{A}$ to register a new party with $\mathsf{sk}_i, \mathsf{pk}_i$ given by $\mathcal{A}$. If party $i$ already exists, then upon this query, all long-term key pairs will be replaced with $\mathsf{sk}_i, \mathsf{pk}_i$, and existing randomness and session keys holding by any $\pi_i^s$ will be erased. In any case, party $i$ has $\tau_i = 0$ once this query has been issued.

- Corrupt($i$): The oracle $\pi_i^s$ responds with the long-term private keys of party $P_i$. If Corrupt($i$) is the $\tau$-th query issued by $\mathcal{A}$, then we say that $P_i$ is $\tau$-corrupted. For parties that have never been corrupted, we define $\tau := \infty$.

- RevealKey($\pi_i^s$): Oracle $\pi_i^s$ responds to this query with the contents of variable $\mathsf{K}_i^s$ to $\mathcal{A}$. This query models the attacks that the exposure of a session key should not be damaging to other sessions. (Note that we have $\mathsf{K}^s \neq \varnothing$ if and only if $\Phi_i^s = \mathtt{accept}$.)

- RevealEph($\pi_i^s$): Oracle $\pi_i^s$ responds with the contents of the ephemeral secret stored in variable $\mathsf{Eph}_i^s$.

- Test($\pi_i^s$): This query can be made at most once. It does not model attacks but functions as a judgment for whether $\mathcal{A}$'s attacks are successful. Oracle $\pi_i^s$ handles this query as follows. If the oracle has state $\Phi_i^s \neq \mathtt{accept}$, then it returns a failure symbol $\bot$. If the oracle does not have access to the corresponding type of keys, it returns some failure symbol $\bot$.

  Otherwise, it flips a fair coin $b$, and it returns $\mathsf{K}_b$, where $\mathsf{K}_0$ is the real $\mathsf{K}_i^s$ and $\mathsf{K}_1 \xleftarrow{\$} \mathcal{K}$.

- TestForge($i, \mathsf{sk}_i', \mathsf{pk}_i'$) This query judges the result of an attack, the goal of which is to forge a valid key pair. The output is 1 if checkKey($\mathsf{sk}_i', \mathsf{pk}_i'$) = TRUE and 0 otherwise, where checkKey() is parameterized by concrete protocols.

### 4.3. Security Definitions

Let $\mathsf{sid}_i^s \in \mathcal{SID}$ denote the session identifier received by oracle $\pi_i^s$, where $\mathcal{SID} \subseteq \{0,1\}^{\mathsf{poly}(\kappa)}$, i.e., a set of binary strings of length $\mathsf{poly}(\kappa)$.

**Definition 8** (Partnering Using sid). *In the protocol execution described above, we say that $\pi_i^s$ (with ($\mathsf{PID}_i^s$, $\Phi_i^s$, $\mathsf{sid}_i^s$)) and $\pi_j^t$ (with ($\mathsf{PID}_t^j$, $\Phi_j^t$, $\mathsf{sid}_j^t$)) are partnered if the following hold for both oracles: (1) $\mathsf{PID}_i^s = j$ and $\mathsf{PID}_t^t = i$; (2) $\Phi_i^s = accept$ and $\Phi_j^t = accept$; (3) $\mathsf{sid}_i^s = \mathsf{sid}_j^t$; .*

**Definition 9** (Registration Freshness). *Let $\mathsf{TestForge}(\mathsf{PID}_i, \mathsf{pk}_i', \mathsf{sk}_i')$ be the $\tau_1$-th query. We call an oracle $\pi_i^s$ $\tau_1$-reg-fresh if all the following conditions hold for the adversary $\mathcal{A}$.*

- *(No direct corruption) $i$ is $\tau$-corrupt with $\tau > \tau_1$.*
- *(No corrupt-and-replace) If $\mathsf{pk}_i' = \mathsf{pk}_j$ and $\mathsf{pk}_j$ is an honest generated public key, then $j$ is $\tau'$-corrupt with $\tau' > \tau_1$.*
- *(Type 1) If the first $\mathsf{RegCorruptParty}(i)$ is the $\tau$-th query with $\tau < \infty$, then $\mathsf{Server}$ is $\tau_S$-corrupt, $\tau_S > \tau_1$, where $\mathsf{Server}$ is the KGC,*
- *(Type 2) If $\mathsf{Corrupt}(\mathsf{Server})$ is the $\tau_S$-th query of $\mathcal{A}$ with $\tau_S < \tau_1$, then $\mathcal{A}$ has not made any $\mathsf{RegCorruptParty}(i)$ before $\tau_S$.*

Here, we define the security of the key pair registration protocol.

**Definition 10** (Secure Key Pair Registration). *Let the KGC be party $\mathsf{S}$. We say that a key pair registration protocol $\Pi$ is $(t, \epsilon)$-secure, if for all adversaries $\mathcal{A}$ with running time bounded by $t$, for some function $\epsilon = \epsilon(\kappa)$, it holds that if $\mathcal{A}$ has issued a $\mathsf{TestForge}(\mathsf{PID}_i, \cdot, \cdot)$-query as the $\tau_1$-th query to oracle $\pi_i^s$, every client oracle $\pi_i^s$ is $\tau_1$-reg-fresh, then the advantage $\mathsf{Adv}_{reg}$ is bounded by a function $\epsilon$. More specifically,*

$$\mathsf{Adv}_{reg} = \Pr[\mathsf{TestForge}(i, \mathsf{pk}_i', \mathsf{sk}_i') = 1 : (i, \mathsf{pk}_i', \mathsf{sk}_i') \xleftarrow{\$} \mathcal{A}^{\mathcal{O}(\cdot)}(1^\kappa)] \leq \epsilon,$$

*where* $\mathcal{O} = \{\mathsf{Send}(), \mathsf{RegCorruptParty}(), \mathsf{Corrupt}()\}$.

**Definition 11** (Session Oracle Freshness). *Let $\pi_i^s$ be an accepting oracle held by a party $P_i$ with intended partner $P_j$. Meanwhile, let $\pi_j^t$ be an oracle (if it exists), such that $\pi_i^s$ and $\pi_j^t$ are partnered. Then the oracle $\pi_i^s$ is said to be $\tau_0$-fresh, if it is $\tau_0$-reg-fresh, and when the adversary $\mathcal{A}$ issues its $\tau_0$-th query to $\pi_i^s$ and NONE of the following conditions holds:*

- *$\mathcal{A}$ has either made a $\mathsf{RevealKey}(\pi_i^s)$ query or a $\mathsf{RevealKey}(\pi_j^t)$ query, (if $\pi_j^t$ exists);*
- *$P_i$ is $\tau_i$-corrupted with $\tau_i \leq \tau_0$;*
- *$P_j$ is $\tau_i$-corrupted with $\tau_j \leq \tau_0$, (if $\pi_j^t$ exists);*
- *if $\pi_j^t$ exists, it is NOT $\tau_0$-reg-fresh;*
- *$\mathcal{A}$ has either made both $\mathsf{RevealEph}(\pi_i^s)$ and $\mathsf{Corrupt}(i)$ queries, or both $\mathsf{RevealEph}(\pi_j^t)$ and $\mathsf{Corrupt}(j)$ (if $\pi_j^t$ exists).*

**Definition 12** (Secure Authenticated Key Exchange). *We say that an AKE protocol $\Pi$ is $(t, \epsilon)$-secure, if for all adversaries $\mathcal{A}$ with running time $t$, for some probability $\epsilon = \epsilon(\kappa)$, it holds that: when $\mathcal{A}$ returns $b'$ such that $\mathcal{A}$ has issued a $\mathsf{Test}()$-query as the $\tau_0$-th query to oracle $\pi_i^s$, and the client oracle $\pi_i^s$ is $\tau_0$-fresh and has a synchronized partner throughout the security game, then the advantage $\mathsf{Adv}_{ake}$ is bounded by a function $\epsilon$. More specifically,*

$$\mathsf{Adv}_{ake} = \left| \Pr[b = b' : b' \leftarrow \mathsf{Adv}^{\mathcal{O}(\cdot)}(1^\kappa)] - 1/2 \right| \leq \epsilon,$$

*where $\mathcal{O} = \{\mathsf{Send}(), \mathsf{RegCorruptParty}(), \mathsf{Corrupt}(), \mathsf{RevealKey}(), \mathsf{RevealEph}(), \mathsf{Test}()\}$.*

The model that we have discussed so far provides a unified framework to evaluate the new CL-AKE protocols. The freshness ensures that we are focusing on real threats, and the queries can be combined to emulate attacks such as replay and man-in-the-middle.

## 5. New Protocols for Certificate-Less Infrastructure

The canonical way to transform a certificate-based AKE to CL-AKE is to replace the signature with a CL-SIG. However, the efficiency of the result is sub-optimal due to redundant computation and the extra demand for randomness. For example, to sign a message with our CL-SIG, an extra randomness is needed for the Schnorr component (line 4 in $\mathsf{Sign}(\mathsf{params}, \mathsf{sk}_i, m)$ in Figure 1), and point multiplications are needed for verification.

Our new CL-AKE protocols save the participants from any signature verification. $\mathsf{Reconst\text{-}Pk}(\mathsf{params}, \mathsf{PID}_i, B_i)$ in $\Pi_{\mathsf{CL\text{-}SIG}}$ (Figure 1) ensures that once Alice has a reliable $\mathsf{pk}_{\mathsf{KGC}} \in \mathsf{params}$, its peer Bob has to use a correct secret key with respect to $B_i$ and $\mathsf{pk}_{\mathsf{KGC}}$ in AKE (Figure 2), leading to optimal performance.

### 5.1. Client Key Registration

Initially, a client is provisioned with the KGC's public keys $\mathsf{ek}_{\mathsf{KGC}}$, $\mathsf{pk}_{\mathsf{KGC}}$ and its own encryption/decryption key pairs. A client can then register its key pair to a KGC, which also knows the client's encryption key. Details of our new client key pair generation protocol (Protocol 2) can be found in Figure 3. The security is summarized in the following theorem, where checkKey for TestForge is defined as checking the discrete log relation $\mathsf{sk}_i' \cdot G = \mathsf{pk}_i'$.

**Theorem 2** (Security of Protocol 2). *Assuming an authenticated channel, if the public encryption scheme $\Pi_{pke}$ is IND-CCA secure and discrete logarithm problem is hard with respect to group $\mathbb{G}$, then Protocol 2 is secure in the sense of Definition 10 in the random oracle model. More specifically, for any PPT adversary $\mathcal{A}_{reg}$,*

$$\mathsf{Adv}_{reg} \leq \frac{(d \cdot \ell)^2}{|\mathbb{G}|} + \epsilon_{\mathsf{H}_1} + \epsilon_{\mathsf{PKE}} + \epsilon_{\mathsf{CL\text{-}SIG}}, \tag{2}$$

*where $d$ is the maximal number of parties, $\ell$ is the maximal number of oracles owned by each party, $\mathbb{G}$ is the group and the range of the hash function $\mathsf{H}()$, $\epsilon_{\mathsf{H}}$ is the advantage against the hash function $\mathsf{H}()$, $\epsilon_{\mathsf{PKE}}$ is the advantage against $\Pi_{\mathsf{PKE}}$ in the IND-CCA game, and $\epsilon_{\mathsf{CL\text{-}SIG}}$ is the advantage against $\Pi_{\mathsf{CL\text{-}SIG}}$.*

**Figure 2.** Protocol 3: Three-pass certificate-less AKE with mutual authentication.

**Proof.** We use a sequence of games [36] to argue $\mathcal{A}$'s advantage against Protocol 2. The term $\mathsf{Adv}_i$ means $\mathcal{A}$'s advantage in $\mathsf{Game}_i$.

$\mathsf{Game}_0$. This is the original game, so we have

$$\mathsf{Adv}_{reg} = \mathsf{Adv}_0 \tag{3}$$

$\mathsf{Game}_1$. We add an abort rule in this game. We abort the game if any collision of honestly generated randomness or any hash collision happens. The abort probability can be bounded by the term $\frac{(d \cdot \ell)^2}{|\mathbb{G}|} + \epsilon_{\mathsf{H}_1}$. Therefore, we have

$$\mathsf{Adv}_0 \leq \mathsf{Adv}_1 + \frac{(d \cdot \ell)^2}{|\mathbb{G}|} + \epsilon_{\mathsf{H}_1}. \tag{4}$$

Once the collisions have all been eliminated, from Corollary 1 we can have

$$\mathsf{Adv}_1 \leq \epsilon_{\mathsf{CL\text{-}SIG}} \tag{5}$$

By combining the (in)equalities (3)–(5), we have (2) in Theorem 2. $\square$

### 5.2. Certificate-Less Authenticated Key Exchange

The certificate-less authenticated key exchange protocol (CL-AKE) with explicit authentication is presented in Figure 2. The correctness is trivial, and the three non-base point multiplications are for computing $\mathsf{pk}_i$ (or $\mathsf{pk}_j$) and ms. Let $\mathsf{H} : \{0,1\}^* \to \mathbb{Z}_{|\mathbb{G}|}$ be a hash function modeled as a random oracle that maps any binary string to an integer in group $\mathbb{Z}_{|\mathbb{G}|}$.

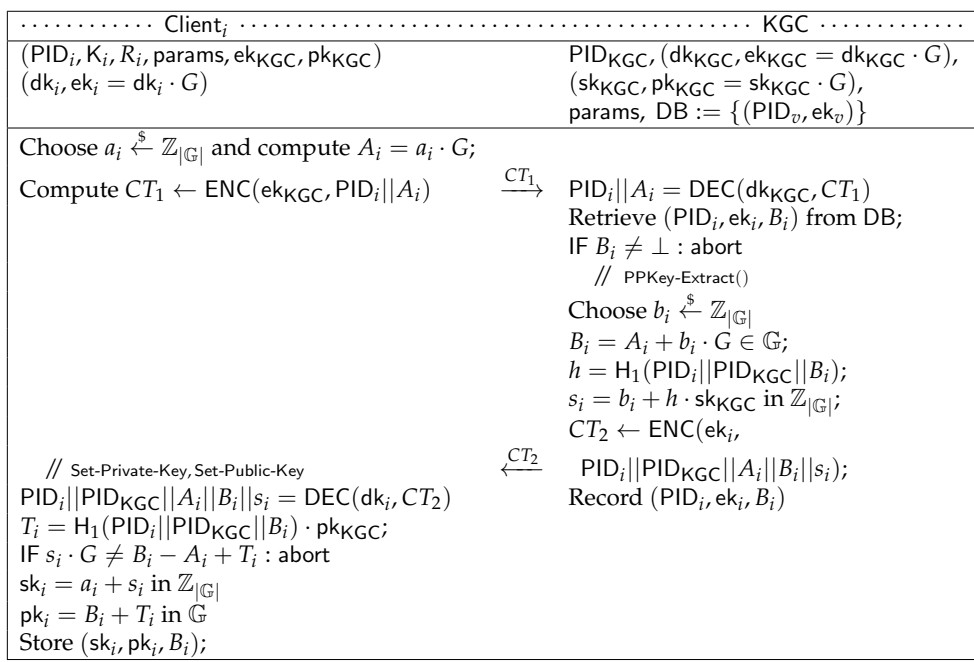

**Figure 3.** Protocol 2, client key pair registration with $\mathsf{msk} = \mathsf{sk}_{\mathsf{KGC}}$.

**Theorem 3** (Security of Protocol 3)**.** *If the key pair registration scheme is secure, the* GCDH *problem is hard, and* PRF *and* MAC *are secure, then Protocol 3 is secure in the sense of Definition* 12 *in the random oracle model. More specifically, for any PPT adversary* $\mathcal{A}_{ake}$,

$$
\mathsf{Adv}_{ake} \leq \mathsf{Adv}_{reg} + \frac{(d \cdot \ell)^2}{|\mathbb{G}|} + \epsilon_{\mathsf{H}_1} + \epsilon_{\mathsf{H}}
$$
$$
+ (d \cdot \ell)^2 \cdot (\epsilon_{\mathsf{PRF}} + \epsilon_{\mathsf{MAC}} + 4 \cdot \epsilon_{\mathsf{GCDH}}), \tag{6}
$$

*where d is the maximal number of parties,* $\ell$ *the maximal number of oracles owned by each party,* $\mathbb{G}$ *is the group for* CL-SIG, $\epsilon_{\mathsf{H}_1}, \epsilon_{\mathsf{H}}$ *the advantages against hash functions* $\mathsf{H}_1()$ *and* $\mathsf{H}()$, $\epsilon_{\mathsf{PRF}}$ *the advantage against the pseudo-random function* PRF, $\epsilon_{\mathsf{MAC}}$ *the advantage against* MAC, *and* $\epsilon_{\mathsf{GCDH}}$ *the advantage against the* GCDH *problem.*

**Proof.** We use another sequence of games to bound $\mathcal{A}$'s advantage against Protocol 3. The term $\mathsf{Adv}_i$ denotes $\mathcal{A}$'s advantage in $\mathsf{Game}_i$.

$\mathsf{Game}_0$. This is the original game, so we have

$$
\mathsf{Adv}_{ake} = \mathsf{Adv}_0 \tag{7}
$$

$\mathsf{Game}_1$. We add an abort rule in this game. If any collision of honestly generated randomness happens, or any hash collision happens, we abort the game. The abort probability can be bounded by the term $\frac{(d \cdot \ell)^2}{|\mathcal{N}|} + \epsilon_{\mathsf{H}_1} + \epsilon_{\mathsf{H}}$, Therefore we have

$$
\mathsf{Adv}_0 \leq \mathsf{Adv}_1 + \frac{(d \cdot \ell)^2}{|\mathcal{N}|} + \epsilon_{\mathsf{H}_1} + \epsilon_{\mathsf{H}}. \tag{8}
$$

$\mathsf{Game}_2$. We add an abort rule here. If $\mathcal{A}$ successfully forges a key pair and uses in the first message in the transportation phase, prior to the corruption of any party, abort the game. This probability is bounded by $\mathsf{Adv}_{reg}$. Thus, we have

$$
\mathsf{Adv}_1 \leq \mathsf{Adv}_2 + \mathsf{Adv}_{reg} \tag{9}
$$

Game₃. We add an abort rule here. Let the challenger first guess $\mathcal{A}$'s target $(i, s)$ and its peer $(j, t)$. If the guess is wrong, abort the game. Thus, we have

$$\mathsf{Adv}_2 \leq (d \cdot \ell)^2 \cdot \mathsf{Adv}_3 \tag{10}$$

Game₄. We replace PRF() with a random oracle RF(). Distinguishing Game₄ from Game₃ implies the existence of another adversary against the security of PRF(). We now have

$$\mathsf{Adv}_3 \leq \mathsf{Adv}_4 + \epsilon_{\mathsf{PRF}} \tag{11}$$

Game₅. We add an abort rule here. If $\mathcal{A}$ successfully forges an MAC.Tag₁ or MAC.Tag₂ in the second or the last message in the transportation phase, prior to the corruption of the target party or its peer, abort the game. This probability is again bounded by $\epsilon_{\mathsf{MAC}}$. Thus, we have

$$\mathsf{Adv}_4 \leq \mathsf{Adv}_5 + \epsilon_{\mathsf{MAC}} \tag{12}$$

We use $\mathsf{fresh}_{..}(\pi_i^s)$ to mark four (sub-)cases when the freshness of $\pi_i^s$ still holds, i.e., $\mathcal{A}$'s attack is non-trivial.

- $\mathsf{fresh}_{LL}(\pi_i^s)$ : $\mathcal{A}$ has never queried both $\mathsf{Corrupt}(i)$ and $\mathsf{Corrupt}(j)$.
- $\mathsf{fresh}_{EE}(\pi_i^s)$ : $\mathcal{A}$ has never queried both $\mathsf{RevealEph}(\pi_i^s)$ and $\mathsf{RevealEph}(\pi_j^t)$.
- $\mathsf{fresh}_{EL}(\pi_i^s)$ : $\mathcal{A}$ has never queried both $\mathsf{RevealEph}(\pi_i^s)$ and $\mathsf{Corrupt}(j)$.
- $\mathsf{fresh}_{LE}(\pi_i^s)$ : $\mathcal{A}$ has never queried both $\mathsf{Corrupt}(i)$ and $\mathsf{RevealEph}(\pi_j^t)$.

It is straightforward to see that if none of the cases exist in Game₅, then $\mathcal{A}$'s attack is trivial. Let $\mathsf{Adv}_5^{\mathsf{fresh}_{..}}$ denote $\mathcal{A}$'s advantage in Game₅ when $\mathsf{fresh}_{..}(\pi_i^s)$ holds. We have from the union bound

$$\mathsf{Adv}_5 \leq \mathsf{Adv}_5^{\mathsf{fresh}_{LL}(\pi_i^s)} + \mathsf{Adv}_5^{\mathsf{fresh}_{EL}(\pi_i^s)} + \mathsf{Adv}_5^{\mathsf{fresh}_{LE}(\pi_i^s)} + \mathsf{Adv}_5^{\mathsf{fresh}_{EE}(\pi_i^s)} \tag{13}$$

We rewrite the computation of ms as

$$\mathsf{ms} = \underbrace{\mathsf{sk}_j \cdot \mathsf{pk}_i}_{LL} + \underbrace{\mathsf{H}(W)^2 y \cdot X}_{EE} + \underbrace{\mathsf{sk}_j \mathsf{H}(W) \cdot X}_{EL} + \underbrace{\mathsf{H}(W) y \cdot \mathsf{pk}_i}_{LE}, \tag{14}$$

Observe that each of the four products on the right-hand side of (14) corresponds to one of the four fresh cases, and each fresh case allows a different strategy of embedding the GCDH. Let Game₅$^{\mathsf{fresh}_{LL}(\pi_i^s)}$ Game₅$^{\mathsf{fresh}_{EE}(\pi_i^s)}$ Game₅$^{\mathsf{fresh}_{EL}(\pi_i^s)}$ and Game₅$^{\mathsf{fresh}_{LE}(\pi_i^s)}$ be the game Game₅ when one of the four cases exist.

Game₅$^{\mathsf{fresh}_{LL}(\pi_i^s)}$. We claim that

$$\mathsf{Adv}_5^{\mathsf{fresh}_{LL}(\pi_i^s)} \leq \epsilon_{\mathsf{GCDH}} \tag{15}$$

We show how to construct a GCDH solver $\mathcal{S}$ to prove (15). $\mathcal{S}$ chooses a random value $K^*$ in the key space and program the random oracle $\mathsf{PRF}(\cdot, W||"\mathsf{MAC}")$ with $K^*$, where $W = B_i||B_j||\mathsf{PID}_i||\mathsf{PID}_j||X||Y||\mathsf{params}_i$ and all the variables are from the target session. Let $(\mathbb{G}, A, B)$ be $\mathcal{S}$'s GCDH challenge. $\mathcal{S}$ sets $(\mathsf{pk}_i, \mathsf{pk}_j)$ to $(A, B)$, aborts the game when

1. $\mathcal{A}$ queries the random oracle with a ms at the place of any PRF queries,
2. and $\mathcal{O}_{DDH}(g, A, B, Z) = \mathsf{TRUE}$ where $Z = \mathsf{ms} - \mathsf{H}(W)^2 x \cdot Y - \mathsf{H}(W) x \cdot \mathsf{pk}_j - \mathsf{H}(W) y \cdot \mathsf{pk}_i$.

In other words, if the game aborts, $\mathcal{S}$ finds a solution to the GCDH instance.

As $\mathsf{fresh}_{LL}(\pi_i^s)$ guarantees that neither $\mathsf{sk}_i$ nor $\mathsf{sk}_j$ would be asked by $\mathcal{A}$, $\mathcal{S}$ can generate all other randomness including $(x, y)$ freely and simulate all the other $\mathsf{Corrupt}()$, $\mathsf{RevealEph}()$

and RevealKey() queries perfectly for $\mathcal{A}$. The probability of aborting the game is thus upper bounded by $\epsilon_{\mathsf{GCDH}}$.

On the other hand, if $\mathsf{Game_5}^{\mathsf{fresh}_{LL}(\pi_i^s)}$ simulated by $\mathcal{S}$ does *not* abort, then $\mathcal{A}$ has never queried the random oracle with the correct value to compute the target session key. Due to the property of random oracle, $\mathcal{A}$ has *zero* advantage in distinguishing the random key $K^*$.

$\mathsf{Game_5}^{\mathsf{fresh}_{EE}(\pi_i^s)}$. $\mathcal{S}$ embeds $(A, B)$ into $(X, Y)$, and aborts when $\mathcal{O}_{DDH}(g, A, B, Z) = \mathsf{TRUE}$ where $Z = \mathsf{H}(W)^{-2} \cdot (\mathsf{ms} - \mathsf{sk}_j \cdot \mathsf{pk}_i - \mathsf{sk}_j\mathsf{H}(W) \cdot X - \mathsf{sk}_i\mathsf{H}(W) \cdot Y)$. This abort means a GCDH solution of $(A, B)$ has been found.

$$\mathsf{Adv_5}^{\mathsf{fresh}_{EE}(\pi_i^s)} \leq \epsilon_{\mathsf{GCDH}} \tag{16}$$

$\mathsf{Game_5}^{\mathsf{fresh}_{EL}(\pi_i^s)}$. $\mathcal{S}$ embeds $(A, B)$ into $(\mathsf{pk}_j, X)$, and aborts when $\mathcal{O}_{DDH}(g, A, B, Z) = \mathsf{TRUE}$ where $Z = \mathsf{H}(W)^{-1} \cdot (\mathsf{ms} - \mathsf{sk}_i \cdot \mathsf{pk}_j - \mathsf{H}(W)^2 y \cdot X - \mathsf{sk}_i\mathsf{H}(W) \cdot Y)$. This abort means a GCDH solution of $(A, B)$ has been found.

$$\mathsf{Adv_5}^{\mathsf{fresh}_{EL}(\pi_i^s)} \leq \epsilon_{\mathsf{GCDH}} \tag{17}$$

$\mathsf{Game_5}^{\mathsf{fresh}_{LE}(\pi_i^s)}$. Similar to the previous case, if $\mathcal{S}$ embeds $(A, B)$ into $(Y, \mathsf{pk}_i)$, it can also find a GCDH solution when the game aborts.

$$\mathsf{Adv_5}^{\mathsf{fresh}_{LE}(\pi_i^s)} \leq \epsilon_{\mathsf{GCDH}} \tag{18}$$

Now, we can conclude from the arguments above and (13) that

$$\mathsf{Adv_5} \leq 4 \cdot \epsilon_{\mathsf{GCDH}}. \tag{19}$$

By combining the (in)equalities (7)–(19), we have proved (6). □

## 6. Integration into TLS and Performance Evaluation

Since our solution is already asymptotically better than other provably secure ones (see Table 1), we only demonstrate its performance in real-world scenarios and compare with the original certificate-based TLS (`TLS_ECDHE_ECDSA_WITH_AES_128_GCM_SHA256`, henceforth **TLS-DHE**).

The two standard ways to integrate the Protocol 3 in Figure 2 are via the certificate type RawPublicKey [37] and via the PSK identities [38] (We merged mTag1 and mTag2 with the finish-messages in the TLS handshake.). When RawPublicKey is chosen, the TLS server sends $\mathbf{M_1}$ in the (Server) Certificate message, and the TLS client sends $\mathbf{M_2}$ in the (Client) Certificate message. When PSK identities is used, the TLS server sends $\mathbf{M_1}$ in the ServerkeyExchange.psk_identity_hint field, and the TLS client sends $\mathbf{M_2}$ in the ClientKeyExchange.psk_identity field.

### 6.1. Set Up

Opting for the PSK identifiers, we insert the encoded PID, the EC group ID and auxiliary information into it. We implement Protocol 3 with the BouncyCastle library and OpenSSL (https://www.bouncycastle.org/ and https://www.openssl.org/, accessed on 8 September 2022) on the server, which runs Ubuntu 18.04.6 on 11th Gen Intel(R) Core(TM) i7-1165G7 @ 2.80 GHz CPU with 16.0 GB RAM. Each client node is emulated with mbedTLS (https://github.com/Mbed-TLS/mbedtls) (accessed on 9 September 2022) on a STM32F107VCT6 (https://www.st.com/resource/en/datasheet/stm32f107 vc.pdf) (accessed on 4 December 2023) board with 32-bit MCU and 64/256 KB Flash, which corresponds to a **Class 2** constrained device [7]. For a fair comparison, we use the widely deployed EC curve NIST P-256 and a two-level certificate chain, i.e., the direct issuer of TLS certificates is trusted by both TLS peers.

*6.2. Results*

6.2.1. Computational Cost

Besides using absolute time, we measured the time consumption for the elementary functions with the base-point multiplication as a unit. We consider only the point multiplication, signature signing and verification to be elementary operations. Here, the cost of point addition, arithmetic operations and hash functions are ignored, as they are relatively negligible compared to the above operations.

We use $t_{BPM}$ to denote the time for computing one base point multiplication (BPM) and use $t_{BPM}$ as a unit.

- A non-base point multiplication (PM) costs 6 $t_{BPM}$. This difference comes from the optimization of base-point multiplication [39].
- A signing costs 2.5 $t_{BPM}$ and verification 8.5 $t_{BPM}$. This 6 $t_{BPM}$ difference comes exactly from the extra non-base point multiplication in the verification. Signing with ECDSA also needs extra operations in the integer group, so it is slower (2.5 $t_{BPM}$) than a simple base-point multiplication (1 $t_{BPM}$).

While the TLS with certificates needs 26.5 $t_{BPM}$, this work needs only 19 $t_{BPM}$, saving at least **28%** local computation time for the cryptographic core. Details are provided in Table 4.

**Table 4.** Computation cost. BPM: base-point scalar multiplication, PM: point scalar multiplication. Columns 2 to 4: number of operations on each side.

| | BPM (1 $t_{BPM}$) | PM (6 $t_{BPM}$) | Sign (2.5 $t_{BPM}$) | Vrfy (8.5 $t_{BPM}$) | Total |
|---|---|---|---|---|---|
| TLS-DHE. | 1 | 1 | 1 | 2 | 26.5 $t_{BPM}$ |
| Protocol 3 TLS | 1 | 3 | 0 | 0 | 19 $t_{BPM}$ |

6.2.2. Communication Cost

In this part, we compare the communication cost between this work and TLS-DHE. We measure the communicated messages using the tool Wireshark (https://www.wireshark.org/) (accessed on 8 September 2022), and count the size of all TLS handshake messages (see Table 5). For one full handshake, TLS-DHE consumes about 2430 bytes , while this work consumes only about 840 bytes. That means our work reduces **65%** of the payload.

**Table 5.** Communication cost comparison in **bytes**. * : Server Response includes Server Hello, Server Certificate, Server Key Exchange, Certificate Request and Server Hello Done.

| Operation | Data$_{TLS-DHE}$ | Data$_{Protocol3}$ |
|---|---|---|
| Client Hello | 309 | 106 |
| Server Response * | 1092 | 421 |
| Client Certificate | 733 | **0** |
| Client Key Exchange | 119 | 266 |
| Certificate Verify | 128 | **0** |
| Change Cipher Spec | 50 | 50 |
| **Total** | ≈2430 | ≈840 (35%) |

6.2.3. Resource Consumption on the Constrained Client

The execution time is 2.09 s for Protocol 3 in the LAN setting (1 Gbps with 0.1 ms latency), saving **70%** of the time compared to TLS-DHE with certificates.

Meanwhile, it is 13.8 KB for TLS-DHE, the maximal RAM consumption during key registration and Protocol 3 is 4.09 KB, making a considerable **70%** reduction in RAM consumption. Whereas TLS-DHE consumes more than 150 KB, the binary of Protocol 3

consumes 48.32 KB in maximum in flash, i.e., a good reduction of **67%** in storage can also be seen.

## 7. Conclusions and Future Work

Without using bilinear pairings, we construct practical certificate-less signature, key registration, and authenticated key exchange protocols with integration to TLS. To the best of our knowledge, our AKE protocols have the lowest number of point-multiplication among DH-based CL-AKE, while enjoying strong security in the eCK model.

We believe that the construction of practical post-quantum-secure CL-AKE can be pursued as meaningful future work, to design general compilers that can transform CL-SIG with two-way-reconstructable PK to CL-AKE, and to analyze the possible equivalence of game-based and universally composable security formalization [4].

**Author Contributions:** Conceptualization, L.D., Y.L. and L.L.; methodology, L.D., Y.L. and L.L.; software, L.L.; formal analysis, L.D. and Y.L.; data curation, L.L.; writing—original draft preparation, L.D.; writing—review and editing, Y.L. and L.L.; visualization, L.D. and L.L. All authors have read and agreed to the published version of the manuscript.

**Funding:** This research received no external funding.

**Data Availability Statement:** Data are contained within the article.

**Conflicts of Interest:** The authors declare no conflict of interest.

## Appendix A. Proof of EUF-CMA Security of $\Pi_{\mathsf{CL\text{-}SIG}}$

*Appendix A.1. Multiple-Forking Lemma*

We recall the multiple-forking lemma introduced by Boldyreva et al. [35].

**Lemma A1** (Multiple-Forking Lemma, Lemma C.5 in [35])**.** *Let $\alpha \in \mathbb{Z}^+$ be a fixed integer. Let $n \geq 1$ be an odd integer and $S$ a set with no less than two elements. Let $\mathcal{B}() : \{0,1\}^* \times S^\alpha \to \mathbb{Z}^2 \times \{0,1\}^*, (x, (s_1, \cdots, s_\alpha)) \mapsto (I, J, \Sigma)$ be a randomized algorithm, where $I$ and $J$ are integers with $0 \leq J \leq I \leq \alpha$. The multiple-forking algorithm $\mathcal{MF}_{\mathcal{B},n}$ associated to $\mathcal{B}$ and $n$ is defined as in Figure A1 where $x \in \{0,1\}^*$.*

*Let* $\mathsf{IGen}$ *be a non-deterministic algorithm that takes no input and returns a binary string. Let*

$$\mathsf{accMf} = \Pr[I \geq 1, \ J \geq 1 : x \xleftarrow{\$} \mathsf{IGen}; (s_1, \cdots, s_\alpha) \xleftarrow{\$} S^\alpha;$$
$$(I, J, \Sigma) \xleftarrow{\$} \mathcal{B}((x, (s_1, \cdots, s_\alpha)));]$$
$$\mathsf{frkMf} = \Pr[b = 1 : x \xleftarrow{\$} \mathsf{IGen}; (b, \mathsf{RessultMF}) \xleftarrow{\$} \mathcal{MF}_{\mathcal{B},n}(x)]$$

*Then*

$$\mathsf{frkMf} \geq \mathsf{accMf} \cdot \left( \frac{\mathsf{accMf}^n}{\alpha^{2n}} - \frac{n}{|S|} \right) \tag{A1}$$

$$\mathsf{accMf} \leq \sqrt[n+1]{\alpha^{2n} \cdot \mathsf{frkMf}} + \sqrt[n+1]{\frac{n \cdot \alpha^{2n}}{|S|}} \tag{A2}$$

This lemma allows forks to happen at two distinct positions $I$ and $J$. When $n = 1$, this lemma collapses to the generalized forking lemma [40], allowing forking at only one position.

```
Algorithm MF_{B,n}(x)
────────────────────────────────────────────────────────────────────
 1 :   Initialize an empty array RessultMF[0, · · · , n];
 2 :   Choose random coins ρ for B, (s_1, · · · , s_α) ←$ S^α;
 3 :   (I, J, Σ_0) ← B((x, (s_1, · · · , s_α)); ρ);
 4 :   if (I = 0 OR J = 0) : return (0, RessultMF);
 5 :   (s_1^1, · · · , s_α^1) ←$ S^α; (I_1, J_1, Σ_1) ← B((x, (s_1^1, · · · , s_α^1)); ρ);
 6 :   if ((I_1, J_1) ≠ (I, J) OR s_I^1 = s_I) : return (0, RessultMF);
 7 :   i ← 2;
 8 :   while (i < n) :
 9 :       (s_1^i, · · · , s_α^i) ←$ S^α; (I_i, J_i, Σ_i) ← B((x, (s_1, · · · s_{J-1}, s_J^i, · · · , s_α^i)); ρ);
10 :       if ((I_i, J_i) ≠ (I, J) OR s_J^i = s_J^{i-1}) : return (0, RessultMF);
11 :       (s_1^{i+1}, · · · , s_α^{i+1}) ←$ S^α;
12 :       (I_{i+1}, J_{i+1}, Σ_{i+1}) ← B((x, (s_1, · · · s_{J-1}, s_J^i, · · · , s_{I-1}^i, s_I^{i+1}, · · · , s_α^i)); ρ);
13 :       if ((I_{i+1}, J_{i+1}) ≠ (I, J) OR s_I^{i+1} = s_I^i) : return (0, RessultMF);
14 :       i ← i + 2;
15 :   for  i = 0 to n :
16 :       RessultMF[i] ← Σ_i;
17 :   return (1, RessultMF; )
```

**Figure A1.** The multiple-forking algorithm $\mathcal{MF}_{\mathcal{B},n}$ associated to $\mathcal{B}, n$ in Lemma A1. We use $\mathcal{MF}_{\mathcal{B},3}$ in the proof of Lemma A2.

*Appendix A.2. Proof of Theorem 1*

Here we prove Theorem 1 via proving Lemma A2 and Lemma A3. We use CL-SIG adversaries to constructing DLP solvers and analyze with Lemma A1.

**Lemma A2** ($\Pi_{\mathsf{CL\text{-}SIG}}$ security against Type 1 adversary)**.** *If there exists an efficient Type I adversary against $\Pi_{\mathsf{CL\text{-}SIG}}$ with advantage $\mathsf{Adv}_{\mathsf{CL\text{-}SIG},1}$, then there exist a forger $\mathcal{S}$ with advantage $\mathsf{Adv}_{\mathsf{SIG}}$ against the Schnorr signature scheme, and a DLP solver $\mathcal{D}$ with advantage $\mathsf{Adv}_{\mathsf{DLP}}$ such that*

$$\mathsf{Adv}_{\mathcal{A},\mathsf{CL\text{-}SIG},1} \leq d \cdot \mathsf{Adv}_{\mathsf{SIG}} + \sqrt[4]{(q_{\mathsf{H}_1} + q_{\mathsf{H}_2})^6 \cdot \mathsf{Adv}_{\mathcal{D},\mathbb{G}}^{\mathsf{DLP}}}$$
$$+ \sqrt[4]{\frac{3 \cdot (q_{\mathsf{H}_1} + q_{\mathsf{H}_2})}{|\mathbb{G}|}} + \frac{d^2 + q_{\mathsf{H}_1}^2 + 2 \cdot q_{\mathsf{H}_2}^2 + 2}{|\mathbb{G}|}, \tag{A3}$$

*where d is the maximal number of clients with distinct identifiers, $q_{\mathsf{H}_1}$ and $q_{\mathsf{H}_2}$ are number of queries to random oracle $\mathsf{H}_1$ and $\mathsf{H}_2$, respectively.*

**Proof.** We define the following events in the CL-SIG security experiments against a Type I adversaries $\mathcal{A}$, i.e., with ReplacePK() but without KGC corruption. Let $\mathsf{Adv}_{\mathcal{A},I,\mathsf{CL\text{-}SIG}}$ be the advantage of $\mathcal{A}$.

- $\mathcal{E}_1$ : $\mathcal{A}$ outputs a forgery $(m, \mathsf{pk}_i, \mathsf{PID}_i, \sigma)$, where $\mathsf{Verify}(\mathsf{params}, \mathsf{pk}_i, \mathsf{PID}_i, \sigma) = \mathsf{TRUE}$, $\mathsf{pk}_i$ has not been replaced, and $m$ has not been queried to the signing oracle $\mathcal{O}(\mathsf{sk}_i, \cdot)$.
- $\mathcal{E}_2$ : $\mathcal{A}$ outputs a forgery $(m, \mathsf{pk}_i', \mathsf{PID}_i, \sigma)$, where $\mathsf{Verify}(\mathsf{params}, \mathsf{pk}_i', \mathsf{PID}_i, \sigma) = \mathsf{TRUE}$, and $\mathsf{pk}_i' \neq \mathsf{pk}_i$ is an adversarial public key for $\mathsf{PID}_i$.

It is easy to see that

$$\mathsf{Adv}_{\mathcal{A},\mathsf{CL\text{-}SIG},1} \leq \Pr[\mathcal{E}_1 \cup \mathcal{E}_2] \leq \Pr[\mathcal{E}_1] + \Pr[\mathcal{E}_2] \tag{A4}$$

To bound $\mathsf{Adv}_{\mathcal{A},I,\mathsf{CL\text{-}SIG}}$, we first prove that

$$\Pr[\mathcal{E}_1] \leq d \cdot \mathsf{Adv}_{\mathsf{SIG}}, \tag{A5}$$

where $d$ is the maximal number of parties, and $\mathsf{Adv_{SIG}}$ is the advantage that any PPT adversary against the Schnorr signature scheme. We construct an adversary $\mathcal{S}$ against Schnorr signature from $\mathcal{A}$. The simulator $\mathcal{S}$ associates its challenge public key $\mathsf{pk}_i$ and the signing oracle $\mathcal{O}(\mathsf{sk}_i, \cdot)$ to party $i$. The master public-secret key $(\mathsf{pk_{KGC}}, \mathsf{sk_{KGC}})$ and all other honest key pairs are generated by $\mathcal{S}$. Signing and other key queries are handled faithfully for all $\mathsf{PID}_j, j \neq i$. For $i$, $\mathcal{S}$ prepares $B_i$, the partial private key $s_i$ and the user secret $a_i$ as follows.

- chooses random $h_i$ in the range of $\mathsf{H}_1$, compute $B_i = \mathsf{pk}_i - h_i \cdot \mathsf{pk_{KGC}}$.
- program the random oracle $\mathsf{H}_1$ such that $\mathsf{H}_1(\mathsf{PID_{KGC}}||\mathsf{PID}_i||B_i) = h_i$.
- If $\mathcal{A}$ queries for $s_i$, randomly choose a $b_i \xleftarrow{\$} \mathbb{Z}_q$ and compute $s_i = b_i + h_i \cdot \mathsf{sk_{KGC}}$.
- If $\mathcal{A}$ queries for $a_i$, randomly choose a $a_i \xleftarrow{\$} \mathbb{Z}_q$.

When $\mathcal{A}$ outputs a forgery $(\mathsf{PID}_i, \mathsf{pk}_i, m, \sigma = (\sigma', B_i))$, $\mathcal{S}$ outputs a forgery $m, \sigma'$ to its own challenger. Note that $\mathcal{A}$ cannot query both $a_i$ and $s_i$, if it does not fail trivially. So the simulation is indistinguishable from a real execution. The loss factor $d$ is from guessing of the challenged party $i$. This completes the proof of (A5).

To bound $\Pr[\mathcal{E}_2]$, first, we define an algorithm $\mathcal{B}$ relying on a $\Pi_{\mathsf{CL\text{-}SIG}}$ forger $\mathcal{A}$ that on inputs $(\text{params}, X)$ and $s_1, \cdots, s_\alpha \in S$, returns a triple $(I, J, \Sigma)$ consisting of two integers $0 \leq J < I \leq \alpha$ and a string $\Sigma$. Details follow.

1. $\mathcal{B}$ gets a DLP challenge $(\mathbb{G}, G, X)$ and a KGC identifier $\mathsf{PID_{KGC}}$.
2. $\mathcal{B}$ sets up the KGC public key $\mathsf{pk_{KGC}} \leftarrow X$, the KGC identifier $\mathsf{PID_{KGC}}$, CL-PKC parameters $\text{params} \leftarrow (\mathbb{G}, G, \mathsf{PID_{KGC}}, \mathsf{pk_{KGC}})$. $\mathcal{B}$ also initializes two empty lists $\mathcal{L}_{\mathsf{H}_1}$ and $\mathcal{L}_{\mathsf{H}_2}$ to simulate the random oracles. Another empty list $\mathcal{L}_{\mathsf{aPK}}$ of replaced public keys is initialized by $\mathcal{B}$. $\mathcal{B}$ set up a flag $\mathsf{bad} \leftarrow \mathsf{FALSE}$.
3. Preparation of simulated signing keys $\{(\mathsf{sk}_j, B_j)\}$ for $\{\mathsf{PID}_j\}_{j=1, j \neq \mathsf{PID_{KGC}}}^d$.

   - Choose random $\mathsf{sk}_j \xleftarrow{\$} \mathbb{Z}_{|\mathbb{G}|}$, compute $\mathsf{pk}_j \leftarrow \mathsf{sk}_j \cdot G$.
   - Choose random $h_j$ in the range of $\mathsf{H}_1()$, compute $B_j = \mathsf{pk}_j - h_j \cdot \mathsf{pk_{KGC}}$. If any $\mathsf{sk}_j$ collision happens, set $\mathsf{bad} = \mathsf{TRUE}$.
   - Program the random oracle $\mathcal{O}_{\mathsf{H}_1}()$ such that $\mathsf{H}_1(\mathsf{PID_{KGC}}||\mathsf{PID}_j||B_j) = h_j$, i.e., set $\mathcal{L}_{\mathsf{H}_1}(\mathsf{PID_{KGC}}||\mathsf{PID}_j||B_j) \leftarrow h_j$.

4. $\mathcal{B}$ sends params to $\mathcal{A}$, chooses some randomness for $\mathcal{A}$, and prepares to answer the random oracle queries and others.
5. Answer to $\mathsf{H}_1(s)$ queries, where $s$ has the form $\mathsf{PID_{KGC}}||\mathsf{PID}_k||B_k$.

   If $\mathcal{L}_{\mathsf{H}_1}(s)$ is defined, return $\mathcal{L}_{\mathsf{H}_1}(s)$ to $\mathcal{A}$. Otherwise, pick up $H \xleftarrow{\$} \mathbb{Z}_{|\mathbb{G}|}$, define $\mathcal{L}_{\mathsf{H}_1}(s) \leftarrow H$, and return $H$.
6. Answer to $\mathsf{H}_2(R||m)$.

   If $\mathcal{L}_{\mathsf{H}_2}(R||m)$ is defined, return $\mathcal{L}_{\mathsf{H}_2}(R||m)$ to $\mathcal{A}$. Otherwise, pick up $W \xleftarrow{\$} \mathbb{Z}_{|\mathbb{G}|}$, define $\mathcal{L}_{\mathsf{H}_1}(R||m) \leftarrow W$, and return $W$.
7. Answer to $\mathsf{PPKey\text{-}Extract}(\mathsf{PID}_j)$

   - If $\mathsf{pk}_j$ is not replaced, choose a $b_j \xleftarrow{\$} \mathbb{Z}_q$, compute $s_j = b_j + h_j \cdot \mathsf{sk_{KGC}}$, record and return $s_j$ to $\mathcal{A}$.
   - Otherwise, return $\bot$.

8. Answer to $\mathsf{ReplacePK}(\mathsf{PID}_j, \mathsf{pk}_j)$ queries. Record $(\mathsf{PID}_j, \mathsf{pk}_j)$ in $\mathcal{L}_{\mathsf{aPK}}$ and return "OK" to $\mathcal{A}$.
9. Answer to $\mathsf{getPrivateKey}(\mathsf{PID}_j)$. Return $\mathsf{sk}_j$ if $\mathsf{pk}_j$ is not replaced, and $\bot$ otherwise.
10. Answer to $\mathsf{SIG.Sign}(\mathsf{PID}_j, m)$ queries.

    - If $(\mathsf{PID}_j, \mathsf{pk}_j) \notin \mathcal{L}_{\mathsf{aPK}}$, retrieve the simulated signing key $\mathsf{sk}_j$.

      – Choose $r \xleftarrow{\$} \mathbb{Z}_{|\mathbb{G}|}$, compute $R \leftarrow r \cdot G$, choose $h_m \xleftarrow{\$} \mathbb{Z}_{|\mathbb{G}|}$.
      – If $\mathcal{L}_{\mathsf{H}_2}(R||m)$ is defined, set $\mathsf{bad} = \mathsf{TRUE}$.
      – Set $\mathcal{L}_{\mathsf{H}_2}(R||m) \leftarrow (h_m, \beta)$, where $\beta = r - h_m \cdot \mathsf{sk}_j$, and return $(R, \beta)$ to $\mathcal{A}$.

- Otherwise, return $\perp$ to $\mathcal{A}$.

We also count signing queries as $\mathsf{H}_2()$ queries.

11. If $\mathcal{A}$ submits a forgery $(\mathsf{PID}_i, \mathsf{pk}_i^*, m^*, \sigma^*)$, $\mathcal{B}$ parse $\sigma^*$ as $(\sigma', B_i^*)$ and parse $\sigma'$ as $(R, \beta)$.

- $\mathcal{B}$ searches for $h_1 = \mathcal{L}_{\mathsf{H}_1}(\mathsf{PID}_i||\mathsf{PID}_{\mathsf{KGC}}||B_i^*)$, and $h_2 = \mathcal{L}_{\mathsf{H}_2}(R||m^*)$. If $h_1$ or $h_2$ is not defined, set bad $=$ TRUE.
- If $\mathsf{pk}_i \neq B_i^* + \mathcal{L}_{\mathsf{H}_1}(\mathsf{PID}_{\mathsf{KGC}}||\mathsf{PID}_i||B_i^*) \cdot X$, set bad $=$ TRUE.

If bad $=$ FALSE, then $\mathcal{B}$ finally outputs $(I, J, \Sigma = \beta||h_1||h_2)$. Otherwise $\mathcal{B}$ outputs $(0, 0, \perp)$.

**Table A1.** Variables in the proof.

| Variable | Meaning |
|---|---|
| $\mathcal{L}_{\mathsf{aPK}}$ | a list of public keys registered by the adversary |
| $\mathcal{L}_{\mathsf{H}_1}$ | a list to simulate the random oracle $\mathsf{H}_1(\cdot)$ |
| $\mathcal{L}_{\mathsf{H}_2}$ | a list to simulate the random oracle $\mathsf{H}_2(\cdot)$ |

We then use the multiple-forking algorithm $\mathcal{MF}_{\mathcal{B},3}$ associated to $\mathcal{B}$ and $n = 3$ to construct a DL solver $\mathcal{D}$ as in Figure A2. Note that we do not require exact $b_i = b_i'$ and $a_i = a_i'$, but $(a_i + b_i) \equiv (a_i' + b_i') \mod \mathbb{Z}_{|\mathbb{G}|}$ for the first fork. This critical forking point is observable by $\mathcal{B}$ in $\mathcal{L}_{\mathsf{H}_1}$ for the $\mathsf{H}_1(\cdots||B_i)$ query, as $B_i = (a_i + b_i) \cdot G$ is unique in $\mathbb{G}$. Similarly, the second fork can be observed for $\mathsf{H}_2()$-queries on $R||m^*$.

---

Algorithm $\mathcal{D}(\mathbb{G}, G, X)$

1: $\mathsf{PID}_{\mathsf{KGC}} \xleftarrow{\$} [d]$;

2: $(b, \mathsf{RessultMF}) \xleftarrow{\$} \mathcal{MF}_{\mathcal{B},3}(\mathbb{G}, G, X, \mathsf{PID}_{\mathsf{KGC}})$;

3: **if** $(b = 0)$ : **return** 0;

4: Parse $\mathsf{RessultMF}[0]$ as $(\beta_0, h_1, h_2)$, $\mathsf{RessultMF}[1]$ as $(\beta_1, h_1', h_2')$,

5: $\quad\quad \mathsf{RessultMF}[2]$ as $(\beta_2, \widetilde{h_1}, \widetilde{h_2})$, $\mathsf{RessultMF}[3]$ as $(\beta_3, \widehat{h_1}, \widehat{h_2})$;

6: **return** $\left((\beta_0 - \beta_1)(h_2 - h_2')^{-1} - (\beta_2 - \beta_3)(\widetilde{h_2} - \widehat{h_2})^{-1}\right)(h_1 - \widetilde{h_1})^{-1} \in \mathbb{Z}_{|\mathbb{G}|}$

---

**Figure A2.** The DLP solving algorithm $\mathcal{D}$ from $\mathcal{MF}_{\mathcal{B},3}$.

We now show that $\mathcal{D}$ solves the DLP for $X$ in $\mathbb{G}$. If $b = 1$, then there exist coins $\rho$ for $\mathcal{B}$, with $j > k \geq 1$ and $s_1, \cdots, s_\alpha, s_j^1, \cdots, s_\alpha^1, s_k^2, \cdots, s_\alpha^2, s_j^3, \cdots, s_{\alpha'}^3, \in \mathbb{Z}_{|\mathbb{G}|}$ with $h_1 = h_1' = s_k \neq s_k^2 = \widetilde{h_1} = \widehat{h_1}$, $h_2 = s_j \neq s_j^1 = h_2'$ and $\widetilde{h_2} = s_j^2 \neq s_j^3 = \widehat{h_2}$. More specifically,

(1) in the execution of $\mathcal{B}(\mathsf{params}, s_1, \cdots, s_\alpha; \rho)$, $\mathcal{A}$ outputs a valid forgery $(\mathsf{PID}_i, pk_i, m, ((R, \beta_0), B_i))$, with $h_1 = \mathcal{L}_{\mathsf{H}_1}(\mathsf{PID}_{\mathsf{KGC}}||\mathsf{PID}_i||B_i) = s_k$, $h_2 = \mathcal{L}_{\mathsf{H}_2}(R||m) = s_j$.

(2) in the execution of $\mathcal{B}(\mathsf{params}, s_1, \cdots, s_{j-1}, s_j^1, \cdots, s_\alpha^1; \rho)$, $\mathcal{A}$ outputs a valid forgery $(\mathsf{PID}_i', pk_i', m', ((R', \beta_1), B_i'))$, with $h_1' = \mathcal{L}_{\mathsf{H}_1}(\mathsf{PID}_{\mathsf{KGC}}||\mathsf{PID}_i||B_i') = s_k$, $h_2' = \mathcal{L}_{\mathsf{H}_2}(R||m) = s_j^1$, $\mathsf{PID}_{\mathsf{KGC}}'||\mathsf{PID}_i'||B_i' = \mathsf{PID}_{\mathsf{KGC}}||\mathsf{PID}_i||B_i$, and $R' = R$.

(3) in the execution of $\mathcal{B}(\mathsf{params}, s_1, \cdots, s_{k-1}, s_k^2, \cdots, s_\alpha^2; \rho)$, $\mathcal{A}$ outputs a valid forgery $(\widetilde{\mathsf{PID}}_i, \widetilde{pk}_i, \widetilde{m}, ((\widetilde{R}, \beta_2), \widetilde{B}_i))$, with $\widetilde{h_1} = \mathcal{L}_{\mathsf{H}_1}(\widetilde{\mathsf{PID}_{\mathsf{KGC}}}||\widetilde{\mathsf{PID}}_i||\widetilde{B}_i) = s_k^2$, $\widetilde{h_2} = \mathcal{L}_{\mathsf{H}_2}(\widetilde{R}||\widetilde{m}) = s_j^2$, and $\widetilde{B}_i = B_i$.

(4) in the execution of $\mathcal{B}(\mathsf{params}, s_1, \cdots, s_{k-1}, s_k^2, \cdots, s_{j-1}^2, s_j^3, \cdots, s_{\alpha'}^3; \rho)$, $\mathcal{A}$ outputs a valid forgery $(\widehat{\mathsf{PID}}_i, \widehat{pk}_i, \widehat{m}, ((\widehat{R}, \beta_2), \widehat{B}_i))$, with $\widehat{h_1} = \mathcal{L}_{\mathsf{H}_1}(\widehat{\mathsf{PID}_{\mathsf{KGC}}}||\widehat{\mathsf{PID}}_i||\widehat{B}_i) = s_k^2$, $\widehat{h_2} = \mathcal{L}_{\mathsf{H}_2}(\widehat{R}||\widehat{m}) = s_j^3$, $\widehat{\mathsf{PID}_{\mathsf{KGC}}}||\widehat{\mathsf{PID}}_i||\widehat{B}_i = \widetilde{\mathsf{PID}_{\mathsf{KGC}}}||\widetilde{\mathsf{PID}}_i||B_i$, and $\widehat{R} = \widetilde{R}$.

From (1) and (2), we have $\mathsf{pk}_i = \mathsf{pk}_i' = B_i + h_1 \cdot X$, $\beta_0 \cdot G = R + h_2 \cdot \mathsf{pk}_i$, and $\beta_1 \cdot G = R + h_2' \cdot \mathsf{pk}_i'$ Since $h_2 \neq h_2'$, $(h_2 - h_2')$ is well-defined. So we have

$$(\beta_0 - \beta_1)(h_2 - h_2')^{-1} \cdot G = B_i + h_1 \cdot X \tag{A6}$$

Similarly, from (3) and (4), we have

$$(\beta_2 - \beta_3)(\widetilde{h_2} - \widehat{h_2})^{-1} \cdot G = B_i + \widetilde{h_1} \cdot X \tag{A7}$$

Solving for $X$ in (A6) and (A7), as $(h_1 - \widetilde{h_1})$ is well-defined, we finally have

$$\left\{ \left( (\beta_0 - \beta_1)(h_2 - h_2')^{-1} - (\beta_2 - \beta_3)(\widetilde{h_2} - \widehat{h_2})^{-1} \right)(h_1 - \widetilde{h_1})^{-1} \right\} \cdot G = X \tag{A8}$$

Therefore, the output of $\mathcal{D}$ when $b = 1$ as in Figure A2 is the DLP solution of X.

Let $q_H = q_{H_1} + q_{H_2}$, where $q_{H_1}$ and $q_{H_2}$ are the number of $\mathcal{A}$'s $H_1()$ and $H_2()$ queries, respectively. Let $\mathsf{bad}|\mathcal{E}_2$ be the event that $\mathcal{B}$ outputs $(0, 0, \perp)$ when $\mathcal{E}_2$ happens. Then we have

$$\Pr[\mathsf{bad}|\mathcal{E}_2] \leq \frac{d^2 + q_{H_1}^2 + 2 \cdot q_{H_2}^2 + 2}{|\mathbb{G}|} \tag{A9}$$

Let IGen be the algorithm that calls $\mathsf{Setup}(1^\kappa)$ for $(\mathsf{params}, \mathsf{msk})$ and returns $\mathsf{params} = (\mathbb{G}, G, \mathsf{pk}_{\mathsf{KGC}}, \mathsf{PID}_{\mathsf{KGC}})$. Let

$$\mathsf{accMf} = \Pr \left[ \begin{array}{l} I \geq 1 \wedge J \geq 1 : \mathsf{params} \xleftarrow{\$} \mathsf{IGen}(1^\kappa); s_1 \cdots s_\alpha \xleftarrow{\$} \mathbb{Z}_{|\mathbb{G}|}; \\ \qquad\qquad (I, J, \sigma) \xleftarrow{\$} \mathcal{B}(\mathsf{params}, s_1 \cdots s_\alpha); \end{array} \right]$$

By applying (A2) in Lemma A1 for $n = 3$ we have

$$\Pr[\mathcal{E}_2] \leq \mathsf{accMf} + \Pr[\mathsf{bad}|\mathcal{E}_2]$$

$$\leq \sqrt[4]{q_H^6 \cdot \mathsf{frkMf}} + \sqrt[4]{\frac{3 \cdot q_H}{|\mathbb{G}|}} + \Pr[\mathsf{bad}|\mathcal{E}_2]$$

$$\leq \sqrt[4]{(q_{H_1} + q_{H_2})^6 \cdot \mathsf{Adv}_{\mathcal{D},\mathbb{G}}^{\mathsf{DLP}}} + \sqrt[4]{\frac{3 \cdot (q_{H_1} + q_{H_2})}{|\mathbb{G}|}} + \Pr[\mathsf{bad}|\mathcal{E}_2], \tag{A10}$$

where $\mathsf{Adv}_{\mathcal{D},\mathbb{G}}^{\mathsf{DLP}}$ is the advantage of $\mathcal{D}$ against DLP in $\mathbb{G}$. By combining (A4), (A5), (A9) and (A10), we get (A3) in Lemma A2. $\square$

**Lemma A3** ($\Pi_{\mathsf{CL\text{-}SIG}}$ against Type II adversary). *If there exists an efficient Type II adversary against $\Pi_{\mathsf{CL\text{-}SIG}}$ with advantage $\mathsf{Adv}_{\mathsf{CL\text{-}SIG},2}$, then there exists a forger $\mathcal{S}$ with advantage $\mathsf{Adv}_{\mathsf{SIG}}$ against the Schnorr signature scheme, such that*

$$\mathsf{Adv}_{\mathcal{A},\mathsf{CL\text{-}SIG},2} \leq d \cdot \mathsf{Adv}_{\mathsf{SIG}} \tag{A11}$$

*where $d$ is the maximal number of clients with distinct identifiers.*

**Proof.** Similar to the previous proof, we define two events such that

$$\mathsf{Adv}_{\mathcal{A},\mathsf{CL\text{-}SIG},2} \leq \Pr[\mathcal{E}_3 \cup \mathcal{E}_4] \leq \Pr[\mathcal{E}_3] + \Pr[\mathcal{E}_4] \tag{A12}$$

- $\mathcal{E}_3$ : $\mathcal{A}$ outputs a forgery $(m, \mathsf{pk}_i, \mathsf{PID}_i, \sigma)$, where $\mathsf{Verify}(\mathsf{params}, \mathsf{pk}_i, \mathsf{PID}_i, \sigma) = \mathsf{TRUE}$, $\mathsf{pk}_i$ is the original public key of party $\mathsf{PID}_i$, and $m$ has not been queried to the signing oracle $\mathcal{O}(\mathsf{sk}_i, \cdot)$.
- $\mathcal{E}_4$ : $\mathcal{A}$ outputs a forgery $(m, \mathsf{pk}_i', \mathsf{PID}_i, \sigma)$, where $\mathsf{Verify}(\mathsf{params}, \mathsf{pk}_i', \mathsf{PID}_i, \sigma) = \mathsf{TRUE}$, and $\mathsf{pk}_i' \neq \mathsf{pk}_i$ is an adversarial public key for $\mathsf{PID}_i$.

If $\mathcal{A}$ does not fail automatically, then we construct an simulator $\mathcal{S}'$ almost identical to $\mathcal{S}$ for event $\mathcal{E}_1$ in Lemma A2, and $\mathcal{S}'$ can answer the extra getKeyKGC() with the simulated msk. So we have

$$\Pr[\mathcal{E}_3] \leq d \cdot \mathsf{Adv}_{\mathsf{SIG}} \tag{A13}$$

The Type 2 adversary cannot replace public keys, so we have $\Pr[\mathcal{E}_4] = 0$. $\quad\square$

We use the advantage $\mathsf{Adv}_{\mathsf{SIG}}$ of attacking Schnorr signature scheme in Theorem 4.1 in [40] to finish the proof.

**Lemma A4** (Security of Schnorr Signature, from Theorem 4.1 in [40]). *If there exist an efficient adversary against the Schnorr signature scheme* $\Pi_{\mathsf{SIG}}$ *with advantage* $\mathsf{Adv}_{\mathsf{SIG}}$*, then there exists a DLP solver* $\mathcal{U}$ *with advantage* $\mathsf{Adv}_{\mathcal{U},\mathsf{DLP}}^{\mathsf{DLP}}$*, such that*

$$\mathsf{Adv}_{\mathsf{SIG}} \leq \sqrt[2]{(q_{\mathsf{H}_2} + q_{\mathsf{SIG}} + 1) \cdot \mathsf{Adv}_{\mathcal{U},\mathbb{G}}^{\mathsf{DLP}}} + \sqrt[2]{\frac{(q_{\mathsf{H}_2} + q_{\mathsf{SIG}} + 1) \cdot (q_{\mathsf{SIG}} + 1)}{|\mathbb{G}|}}, \tag{A14}$$

*where* $q_{\mathsf{H}_2}$ *is the number of random oracle queries for hash function* $\mathsf{H}_2 : \{0,1\}^* \to \mathbb{Z}_{|\mathbb{G}|}$*, and* $q_{\mathsf{SIG}}$ *the number of signing queries.*

By plugging (A14) into (A3) and (A11) and using union bound, we finally arrive at Theorem 1.

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
