# Peer review of "Practical Certificate-Less Infrastructure with Application in TLS"

_cryptography, doi:10.3390/cryptography7040063_

Round 1

Reviewer 1 Report

Comments and Suggestions for Authors

1. Before introducing the new CL-AKE protocol, it is essential to present a comprehensive system model that encompasses all key participants and their operations. Providing an overview of the proposed protocol will help readers understand its contribution to the field.

2. In Section 6.2, the computational cost is compared based on the parameter BPM, which is meaningful as base point multiplication is crucial in elliptic curve operations. However, the definition of the respective costs for different computations is lacking. For instance, the costs of point multiplication, signing, and verification are specified as 6 tBMP, 2.5 tBPM, and 8.5 tBMP, respectively.

3. While the study compares TLS-DHE with the proposed protocol, it is important to note that there are several similar relevant works in existence. It is recommended to broaden the scope of the comparison by including more approaches to emphasize the contribution of this study.

Comments on the Quality of English Language

Minor editing of English language is suggested.

Reviewer 2 Report

Comments and Suggestions for Authors

Notes about the manuscript
Title: "Practical Certificate-less Infrastructure with Application in TLS"
ID: cryptography-2701500
Journal: CRYPTOGRAPHY MDPI.

The authors propose certificate-less (CL) protocols for the infrastructure used by authenticated key exchange (AKE). Their constructions is based on elliptic curves without paring, claiming that it can be easily supported by most industrial cryptography libraries on contrained devices. They also claim that their proposal when compared with other paring-free certificate-less solutions enjoys the least number of scalar mutiplicatrion over elliptic curves groups. They use a unified game-based odel to formalize the security of each protocol. They claim that an integration of the core protocols into TLS cipher suites is presented as a stand-alone implementation for contrained devices.

The paper includes the following sections:

Summary
Keywords
1. Introduction
1.1. Our contribution and paper outline
1.2. Related works
1.2.1. IBC and ABC-based CL Solutions.
1.2.2. CL-PKC and Pairing-based Attempts.
1.2.3. Pairing-free CL-AKE.

2. Notation and preliminaries.
2.1. Cryptographic Primitives and Hardness Assumptions

3. New Certificate-less Signature with Two-way Reconstructable Public Key

4. Game-based Security Model for CL-AKE
4.1. Protocol Execution Environment
4.2. Adversary Model
4.3. Security Definitions

5. New Protocols for Certificate-less Infrastructure
5.1. Client Key Registration
5.2. Certificate-less Authenticated Key Exchange

6. Integration into TLS and Performance Evaluation
6.1. Set up
6.2. Results 445
6.2.1. Computational Cost
6.2.2. Communication Cost.
6.2.3. Resources Consumption on the Constrained Client.

7. Conclusion and Future Work

Author Contributions
Funding
Institutional Review Board Statement
Informed Consent Statement
Data Availability Statement
Conflicts of Interest
Sample Availability

Appendix A. Proof of EUF-CMA Security of ΠCL-SIG
References

Comments:

1. It is recommended that the authors proofread the writing and spelling of the whole document. English language writing should be carefully reviewed.
Examples:
- ... still widely deployed, Even in relatively new ...
- ... TLS ciphersuites ...

2. Authors use CL-PKE instead CL-AKE (or CL AKE), or CL-PKE has not been defined (see line 26).

3. Authors should define CL-SIG (see line 26), CL-PKE/SIG (see line 38), and KGC (se line 39).

4. Authenticated key exchange protocols (AKE), a certificate-less (CL), and elliptic-curves (EC) are defined many times (see lines 2, 13, and 46; see lines 1, 50; see lines 2, 52).

5. Authors should define all variables and acronyms before using them.

6. Authors should include recent works in Section "Related works".

7. In Definition 4, authors should define the mentioned algorithms. Some of them are described in Table 2. Definition 4 refers to eigth algorithms but only seven are considered in the list and five in Table 2.

8. Table 2 and Table A1 have the same information.

9. The definitions of Type I and II adversaries in lines 205 to 207 do not match the restrictions in lines 209 to 219.

10. The authors do not explain the foundations that support the first construction of their proposal. All variables must be defined in Fig. 1 and Eq. 1. They should consider a diagram to explain the integration of their proposal.

11. The authors do not describe the methods to develop their proposal.

12. In Section 3, authors do not explain why their proposal is with two-way reconstructable public key. In Table 2, "Reconst-Pk" does not considered and it is a one-way function.

13. Authors should explain the reasons for proposing the indicated extensions to the eCK model. What vulnerabilities or flaws did they find in the eCK model to propose the indicated extensions?. The real proposal is in these extensions, but the authors do not explain how they developed and defined them, nor did they indicate the deficiencies of the original algorithm that led to include them in this scheme.

14. Authors should explain the importance of Section 4.3 in terms of future Sections of their manuscript.

15. Authors should include a protocol diagram to explain the functionality of their porposal.

16. Why the efficiency of the result is sub-optimal in the canonical transformation from certificated-based AKE to CL-AKE? See line 328. Where are the redundant computations and extra demand for randomness?

17. The problem addressed in this manuscript is not precisely identified and defined.

18. The way this manuscript is presented is very unfortunate, the authors do not present all the considerations that they make to develop the proposed new scheme.

19. This manuscript does not present an in-depth analysis of the scheme from which they start to show the need to incorporate extensions for complementing the functionality of a CL-AKE applicable to constricted devices.

20. In Fig. 3, the authors mention Protocol 2 but they do not mention the Protocol 1 before. Protocol 2 has an explanatory diagram, but Protocol 3 does not have it.

21. Table 1 shows the comparison of this proposal against other models. Section 6 refers to Table 1, but the claims are not proven. Is the solution asymptotically better than other provably secure ones?

22. This manuscript does not provide information to demonstrate that the results presented are reproducible and repeatable. The text shows links to other parts of the manuscript, and from those to other parts, but does not make reliable demonstrations of the claims made by the authors.

Comments on the Quality of English Language

It is recommended that the authors proofread the writing and spelling of the whole document. English language writing should be carefully reviewed.
Examples:
- ... still widely deployed, Even in relatively new ...
- ... TLS ciphersuites ...

There are many errors.

Reviewer 3 Report

Comments and Suggestions for Authors

The paper introduces some novel concepts, particularly in the context of certificate-less infrastructure in TLS. However, the overall novelty is somewhat limited as the core concepts are not entirely new (or authors need to better/clearer explain novelty and/or their contribution). Without doubts, the content is highly significant, addressing crucial aspects of network security and TLS in the context of evolving cybersecurity challenges. Its relevance to IoT and post-quantum security is especially notable. The paper targets a specialized audience interested in cryptography, TLS, and network security. While its technical depth is appealing to specialists, it may not attract a broader audience due to some crucial issues. Below I summarize key shortcomings in each evaluated category.

Introduction and Background Sufficientness:
The introduction provides a solid technical foundation, focusing on AKE protocols and CL-AKE evolution. However, it lacks broader context, particularly in IoT security challenges and TLS implications. There's an assumption of prior knowledge, which could be limiting for readers outside the immediate field. Simpler/shorter sentences could for sure help to reader orient in the very field-specific technical paper together with broader explanation of findings with its highlighting.

Relevance of Cited References: References cited are generally relevant, showcasing a good grasp of existing literature in CL-AKE and cryptographic protocols. No recommendations here.

Appropriateness of Research Design: The research design is methodical and appears appropriate focusing on secure protocol development and transformations in cryptographic frameworks. However, details on validation against existing standards and practical applicability, particularly in TLS contexts, are lacking and could help with clearer applicability.

Adequacy of Methods Description: No doubts, the paper is scientifically sound, with a rigorous approach and thorough analysis. It provides clear expositions of cryptographic principles and security analyses. Methods, especially cryptographic cores, algorithms, and mathematical proofs, are often described with rigor but lack accessibility for readers not deeply versed in the field. Details and justifications for specific choices are frequently missing. This might greatly broaden the audience interest and understanding.

Clarity of Results Presentation: Results, where presented, are often entangled with technical details, reducing clarity. Clearer, more intuitive presentations, separate from complex mathematical proofs, would be beneficial for broader audience. Simplification, better structuring, and the inclusion of visual aids would significantly enhance the paper's readability and accessibility.

Support of Conclusions by Results: In sections where conclusions are drawn, they appear to be supported by detailed proofs and analyses. However, the connection between results and conclusions is not always explicit, and in some cases. Due to the length, the paper would benefit from highlighting sub-conclusions in each chapter.

Overall Evaluation: The paper presents a valuable contribution to network security and cryptography, particularly in the context of IoT. However, it requires some improvements in methodological detailing, clarity of presentation, structure and contextualization. The technical rigor is apparent, but accessibility and practical applicability need to be enhanced to realize the full potential of the research.

Comments on the Quality of English Language

The English is generally clear across the chapters, but there are recurring instances of complex sentence structures and minor grammatical errors. Simplification and minor editing would enhance clarity, especially for non-expert readers.

Round 2

Reviewer 2 Report

Comments and Suggestions for Authors

Notes about the manuscript
Title: "Practical Certificate-less Infrastructure with Application in TLS"
ID: cryptography-2701500 V2
Journal: CRYPTOGRAPHY MDPI.

The authors propose certificate-less (CL) protocols for the infrastructure used by authenticated key exchange (AKE). Their constructions is based on elliptic curves without paring, claiming that it can be easily supported by most industrial cryptography libraries on contrained devices. They also claim that their proposal when compared with other paring-free certificate-less solutions enjoys the least number of scalar mutiplicatrion over elliptic curves groups. They use a unified game-based odel to formalize the security of each protocol. They claim that an integration of the core protocols into TLS cipher suites is presented as a stand-alone implementation for contrained devices.

The paper includes the following sections:

Abstract
Keywords
1. Introduction
1.1. Our contribution and paper outline
1.2. Technical road map
1.3. Related works
1.3.1. IBC and ABC-based CL Solutions.
1.3.2. CL-PKC and Pairing-based Attempts.
1.3.3. Pairing-free CL-AKE.

2. Notation and preliminaries.
2.1. Notations
2.1. Cryptographic Primitives and Hardness Assumptions

3. New Certificate-less Signature with Two-way Reconstructable Public Key

4. Game-based Security Model for CL-AKE
4.1. Protocol Execution Environment
4.2. Adversary Model
4.3. Security Definitions

5. New Protocols for Certificate-less Infrastructure
5.1. Client Key Registration
5.2. Certificate-less Authenticated Key Exchange

6. Integration into TLS and Performance Evaluation
6.1. Set up
6.2. Results
6.2.1. Computational Cost
6.2.2. Communication Cost.
6.2.3. Resources Consumption on the Constrained Client.

7. Conclusion and Future Work

Author Contributions
Funding
Institutional Review Board Statement
Informed Consent Statement
Data Availability Statement
Conflicts of Interest
Sample Availability

Appendix A. Proof of EUF-CMA Security of ΠCL-SIG
References

Comments:

1. Ok

2. Ok

3. Ok

4. Ok

5. Ok

6. Ok

7. Ok

8. Ok

9. Ok

10. Ok

11. Ok

12. Ok

13. Ok

14. Ok

15. Ok

16. Ok

17. Ok

18. Ok

19. Ok

20. Ok.

21. Ok

22. Ok